# Adversarially Robust Change Point Detection

**Mengchu Li**
Department of Statistics
University of Warwick
mengchu.li@warwick.ac.uk

**Yi Yu**
Department of Statistics
University of Warwick
yi.yu.2@warwick.ac.uk

## Abstract

Change point detection is becoming increasingly popular in many application areas. On one hand, most of the theoretically-justified methods are investigated in an ideal setting without model violations, or merely robust against identical heavy-tailed noise distribution across time and/or against isolate outliers; on the other hand, we are aware that there have been exponentially growing attacks from adversaries, who may pose systematic contamination on data to purposely create spurious change points or disguise true change points. In light of the timely need of a change point detection method that is robust against adversaries, we start with, arguably, the simplest univariate mean change point detection problem. The adversarial attacks are formulated through the Huber $\varepsilon$-contamination framework, which in particular allows the contamination distributions to be different at each time point. In this paper, we demonstrate a phase transition phenomenon in change point detection. This detection boundary is a function of the contamination proportion $\varepsilon$ and is the first time shown in the literature. In addition, we derive the minimax-rate optimal localisation error rate, quantifying the cost of accuracy in terms of the contamination proportion. We propose a computationally-feasible method, matching the minimax lower bound under certain conditions, saving for logarithmic factors. Extensive numerical experiments are conducted with comparisons to existing robust change point detection methods.

## 1 Introduction

Change point detection is attracting tremendous attention due to the demand from various application areas, including bioinformatics [e.g. 1, 2], climatology [e.g. 3, 4] and finance [e.g. 5, 6], among many others. In the last few decades, a vast body of methods and theory on change point analysis have been studied based on different data types [e.g. 7, 8, 9, 10, 11, 12]. The majority of the methods are studied in a model-specific way, in the sense that the noise distributions are sub-Gaussian/sub-Exponential and the between change points data are independent and identically distributed. The few robust change point detection results [e.g. 13, 14, 15] are designed against isolated outliers and/or heavy-tailed noise.

In recent years, we are aware of the risk of adversary attacks in emerging application areas, ranging from image classification [16], object detection [17], to natural language processing [18] and beyond. In this paper, we consider a change point analysis setting possibly attacked by adversaries. As a teaser, we consider a concrete climate data set from [19] containing the daily average PM2.5 index data in Beijing from 15-Apr-2017 to 15-Feb-2021. The original and adversarially contaminated data (with a spurious change point created on 17-Jan-2020) are shown in Figure 1, where the orange points in the right panel denote the adversarially chosen contamination points. As we will see more details in Sections 4.2 and E.3, applying the BIWEIGHT(3) method [15] to the original data set leads to two change point estimators, while only the spurious change point is detected in the presence of

35th Conference on Neural Information Processing Systems (NeurIPS 2021).

contamination. In contrast, our proposed method aARC is less affected by the adversarial attack and detects the same two change points with and without the presence of the contamination.

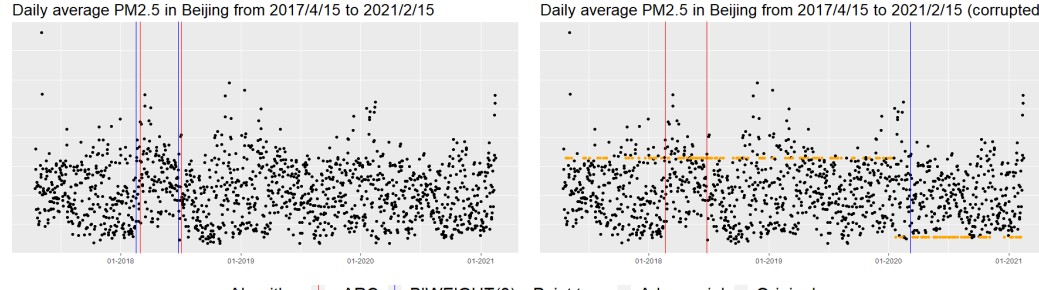

Figure 1: Adversarial attacks on a PM2.5 index data set. Red and blue lines indicate the change points estimators of aARC and BIWEIGHT(3); lines in the left panel are jittered to be visible.

From this example, we see that the presence of adversarial contamination can significantly affect the performance of even the state-of-the-art robust change point detection algorithm and lead to detecting spurious change points, while missing out the ones that can be discovered without contamination. In general, three effects of the adversarial attacks are of interest: **(i)** hiding true change points; **(ii)** creating spurious change points; and **(iii)** increasing the localisation error rate.

To armour the change point detection procedure against potential adversarial attacks, in this paper, we establish and investigate a univariate mean change point detection framework, under a dynamic extension of the Huber $\varepsilon$-contamination model (1). Our contributions in this paper are threefold.

• To the best of our knowledge, this is the first paper formalising the change point detection framework with the dynamic Huber $\varepsilon$-contamination model (2). To be specific, the contamination distributions are allowed to be distinct at every time point. Most if not all of the robust change point detection papers, despite that they have shown their methods are robust against outliers and/or heavy-tail noise, the theoretical framework they study are still within the i.i.d. territory. This has handicapped the existing work to study the adversary attacks, where adversaries may design specific contamination based on their knowledge of the underlying models.

• In Section 2, we propose a signal-to-noise ratio quantity $\kappa/\sigma$ and show that in the regime $\kappa/\sigma \lesssim \sqrt{\max\{\varepsilon, \log(n)/L\}}$, no algorithm is guaranteed to estimate the change points consistently in the sense of (3); and we show in Section 3 that in the regime $\kappa/\sigma \gtrsim \sqrt{\max\{\varepsilon, \log(n)/L\}}$ and $L \gtrsim \log(n)$, our proposed algorithm can localise change points consistently with properly chosen tuning parameters. The localisation error rate can also be nearly minimax optimal, off by a logarithmic factor, for a certain range of model parameters. The detection boundary matches that in the standard change point literature (i.e. $\varepsilon = 0$) and that in the robust statistics literature (i.e. $\kappa = 0$). Compared to the standard literature in the univariate mean change point detection [20, 21], our results can quantify the cost of the contamination in terms of $\varepsilon$, and shed light on both the cost of robustness and designs of adversarial attacks.

• The Adversarially Robust Change point detection algorithm (ARC) that we propose in Algorithm 1 is a combination of [22] and a simple scanning idea. Prasad et al. [22] showed that their robust univariate mean estimator (RUME) can provide optimal estimation without the presence of change points. The scanning idea has been widely used with numerous variants, but none of which is studied in an adversarial setting. In our paper, we exploit the potential of these two areas and investigate both the theoretical and numerical performances of ARC. In addition, we also propose a variant of ARC, namely automatic ARC (aARC), which adapts to the contamination proportion $\varepsilon$.

## 1.1 Related literature

Without the concern of robustness, the theoretical framework of univariate mean change point analysis problem is well established [e.g. 21, 20], with a host of algorithms available to practitioners, including penalised least square methods [e.g. 9] and CUSUM-based methods [e.g. 23], among many others.

When the robustness comes into play, a line of attack has been deployed recently. Fearnhead and Rigaill [15] considered a penalised $M$-estimation procedure, which is designed particularly against outliers with large variances. Yu and Chen [24] studied a testing problem, utilising a $U$-statistic-type test statistics and showing that it is robust against i.i.d. heavy-tailed noise distributions. Hušková [25] and Hušková [13] adapted Huber's theory on robust statistics to study a regression change point detection problem. There has also been work on the online version of the robust change point detection problem [e.g. 26].

In the robust statistics literature, without the presence of change points, the heavy-tailed model and the Huber $\varepsilon$-contamination model are the two in the spotlight. A fundamental problem therein is to estimate the mean of the underlying distribution or the decontaminated distribution accurately. Efficient and optimal algorithms have been developed for both models separately [e.g. 27, 28]. Some work tackles the two models simultaneously. For instance, Prasad et al. [22] exploited the connection between these two models and developed a computationally-efficient univariate mean estimator that is optimal under both models. Hopkins et al. [29] and Diakonikolas et al. [30] considered a filter-type algorithm, which was developed under the high-dimensional contamination model, and showed that it achieves optimal error guarantees under the heavy-tailed model as well. It is worth noting that the theoretical results derived in this paper is not a straightforward adaptation of the existing literature due to the presence of potentially multiple change points.

## 1.2 Problem setup

We kick off the formalisation of the problem with the Huber $\varepsilon$-contamination model [31]

$$F_\varepsilon = (1 - \varepsilon)F + \varepsilon H, \tag{1}$$

where $F$ and $H$ are the distributions of interest and arbitrary contamination, respectively, and $\varepsilon$ measures the strength of contamination. This model is widely used in the robust statistics literature, but usually if not always, it is assumed that $F$ is *sub-Gaussian* or has some form of symmetry, and the data are *i.i.d.* from $F_\varepsilon$ [e.g. 32, 33, 22]. In our model assumption below, we relax both restrictions by considering $F$ to be potentially heavy-tailed and allowing $H$ to vary across time, which we refer to as the *dynamic Huber $\varepsilon$-contamination model*, and which prompts it to model the three types of adversarial attack we mentioned previously, as well as a wide range of less adversarial attacks studied already in the literature.

**Assumption 1.** *Let $\{Y_i\}_{i=1}^n \in \mathbb{R}$ be a sequence of random variables with distributions*

$$(1 - \varepsilon_i)F_i + \varepsilon_i H_i, \quad i \in \{1, \ldots, n\}, \tag{2}$$

*where $F_i$'s are distributions with means $f_i$'s and variances upper bounded by $\sigma^2 < \infty$, $H_i$'s are the distributions of arbitrary contamination and $\varepsilon_i$'s are the proportions of contamination upper bounded by $\varepsilon \in (0, 1/2)$. Let $\{\eta_k\}_{k=0}^{K+1} \subset [0, n]$ be a strictly increasing integer sequence with $\eta_0 = 0$, $\eta_{K+1} = n$, satisfying that $f_{t+1} \neq f_t$, if and only if $t \in \{\eta_k\}_{k=1}^K$. Further assume that $F_{\eta_k+1} = \ldots = F_{\eta_{k+1}}$ for $k = 0, \ldots, K$. Let the minimal spacing $L$ and jump size $\kappa$ be*
$$L = \min_{k=0}^K \{\eta_{k+1} - \eta_k\} \text{ and } \kappa = \min_{k=1}^K \kappa_k = \min_{k=1}^K |f_{\eta_k+1} - f_{\eta_k}|.$$

With Assumption 1, our goal can be formalised as detecting any change on $f_i$'s, provided that $\kappa > 0$, in the presence of adversarial noise $H_i$'s. In some applications, adversaries may have access to the generating process (2), and in particular have control over $H_i$'s. As a consequence, adversaries may be able to design attacks creating spurious change points or cancelling out the change point patterns in $F_i$'s. We are the first attempt in discussing the robust change point detection, allowing for such form of structural attacks.

We aim to obtain consistent change point estimators $\{\hat{\eta}_k\}_{k=1}^{\hat{K}}$ such that with high probability it holds that

$$\widehat{K} = K \quad \text{and} \quad n^{-1} d_{\mathrm{H}} \left( \{\eta_i\}_{i=1}^K, \{\widehat{\eta}_i\}_{i=1}^{\widehat{K}} \right) \to 0, \tag{3}$$

where $d_{\mathrm{H}}(\cdot, \cdot)$ is the two-sided Hausdorff distance (see Definition A.1 in the supplementary material).

In the case when $\kappa = 0$, i.e. there is no change point on the signal $f_1, \ldots, f_n$, regardless of the situations of $H_i$'s, we would like to have

$$\mathbb{P}(\widehat{K} = 0) \to 1.$$

To highlight that $F_i$'s are the distributions of interest, in Assumption 1, we restrict the power of an adversary by assuming $\varepsilon < 1/2$. It is apparent that when $\varepsilon \geq 1/2$, detecting the change points in $F_i$'s may be impossible regardless of any other model parameters.

## 2 Phase Transition and Minimax Lower bounds

Without contamination, it is well established that, if $\kappa\sqrt{L}\sigma^{-1} \lesssim \sqrt{\log(n)}$, then in the minimax sense, no algorithm is guaranteed to produce consistent estimators; if $\kappa\sqrt{L}\sigma^{-1} \gtrsim \sqrt{\log(n)}$, then the optimal localisation rate is of order $\sigma^2\kappa^{-2}$ [e.g. 21, 20]. In this section, we are to present the counterparts of such results in the model described in Assumption 1. In detail, Lemma 1 shows that if

$$\kappa/\sigma \lesssim \sqrt{\max\{\varepsilon, \log(n)/L\}}, \tag{4}$$

then no consistent estimator exists. In particular, by considering the two regimes inherited in (4), we can identify two sources of difficulties in detecting change points in an adversarial setting.

**Small $\varepsilon$ regime**. When $\varepsilon \lesssim \log(n)/L$, condition (4) is reduced to $\kappa\sigma^{-1} \lesssim \sqrt{\log(n)/L}$, which is essentially the boundary without the presence of contamination (cf. Section 2 in [21]). An interesting interpretation of the result is that it quantifies how much contamination will affect the fundamental difficulty of the problem and this threshold is of order $\log(n)/L$. Similar phenomena have been observed in the robust mean estimation [e.g. 34] and testing [e.g. 35] problems.

**Large $\varepsilon$ regime**. When $\varepsilon \gtrsim \log(n)/L$, condition (4) is reduced to $\kappa/\sigma \lesssim \sqrt{\varepsilon}$. The term $\sigma\sqrt{\varepsilon}$ is essentially the minimal asymptotic bias that any estimator of the means $f_i$'s must suffer under the Huber $\varepsilon$-contamination model (1) with finite variance $\sigma^2$ [36, 22]. Here, we assert that if the jump size $\kappa$ is no larger than the asymptotic bias $\sigma\sqrt{\varepsilon}$, then no consistent estimator exists. Referring to the three attack strategies we mentioned in Section 1, this is the situation where the signal of a change point can be completely hidden by contamination.

**Lemma 1.** *Let $\{Y_i\}_{i=1}^n$ satisfy Assumption 1. Suppose that $\varepsilon_i = \varepsilon$, $i = 1, \ldots, n$. Let $P_{\kappa,L,\sigma,\varepsilon}^n$ denote the corresponding joint distribution. Consider the class of distributions*

$$\mathcal{P} = \left\{P_{\kappa,L,\sigma,\varepsilon}^n : \kappa^2 L \sigma^{-2} < \max\{8\varepsilon/(1-2\varepsilon), \log(n), 4\varepsilon L\}, L \leq \lfloor n/4 \rfloor\right\}.$$

*For all $n \in \mathbb{N}$ large enough, it holds that $\inf_{\hat{\eta}} \sup_{P \in \mathcal{P}} \mathbb{E}_P\{d_H(\hat{\eta}, \eta(P))\} \geq n/8$, where the infimum is over all possible measurable functions of the data and $\eta(P)$ is the set of true change points of $P \in \mathcal{P}$.*

Note that in many interesting scenarios, $\log(n) \geq 8\varepsilon/(1-2\varepsilon)$, which is equivalent to $\varepsilon < \log(n)/\{2\log(n) + 8\}$. Therefore, for simplicity, in the following, we only consider the condition $\kappa/\sigma \lesssim \max\{\sqrt{\varepsilon}, \sqrt{\log(n)/L}\}$, under which, no algorithm is ensured to output a consistent estimator. As we will show later in Theorem 1, ARC can produce consistent estimation under nearly optimal conditions.

Our second task is to demonstrate the optimal localisation error, under a higher signal-to-noise ratio condition. To be specific, we consider

$$\min\left\{\kappa^2 L \sigma^{-2}, (1-2\varepsilon)\log\{(1-\varepsilon)/\varepsilon\} L\right\} \geq \zeta_n, \tag{5}$$

with $\{\zeta_n\}$ being an arbitrarily diverging sequence. Noting that $x \mapsto (1-2x)\log\{(1-x)/x\}$ is a decreasing function on $(0, 0.5]$, to see how (5) complements (4), we consider the following two regimes.

**Small $\varepsilon$ regime**. If $\kappa^2\sigma^{-2} < (1-2\varepsilon)\log\{(1-\varepsilon)/\varepsilon\}$, then (5) is $\kappa^2 L\sigma^{-2} \geq \zeta_n$, which complements the small $\varepsilon$ regime implied by (4), and under which, as implied by Lemma 2, the lower bound on the localisation rate is of order $\sigma^2\kappa^{-2}$. This is the same rate without the presence of contamination and this again quantifies the cost of contamination, that is to say, if $\varepsilon$ is small enough, one can hope for a localisation error as if there is no contamination.

**Large $\varepsilon$ regime**. If $\kappa^2\sigma^{-2} \geq (1-2\varepsilon)\log\{(1-\varepsilon)/\varepsilon\}$, then (5) is $(1-2\varepsilon)\log\{(1-\varepsilon)/\varepsilon\} L \geq \zeta_n$. This complements the large $\varepsilon$ regime implied by (4). The corresponding lower bound, as implied by Lemma 2, will diverge if $\varepsilon$ tends to $1/2$ at an arbitrary rate. If $\varepsilon$ is bounded away from $1/2$, then regardless of the strength of $\kappa$, the lower bound is of constant order, which is in fact trivial due to the discrete nature of the change points.

**Lemma 2.** *Let $\{Y_i\}_{i=1}^n$ satisfy Assumption 1 with only one change point and let $P_{\kappa,L,\sigma,\varepsilon}^n$ denote the corresponding joint distribution. Suppose that $\varepsilon_i = \varepsilon$, $i = 1, \ldots, n$. Consider the class of distribution*

$$\mathcal{P} = \left\{ P_{\kappa,L,\sigma,\varepsilon}^n : \, \min\left\{ \kappa^2 L \sigma^{-2}, \, (1-2\varepsilon)\log\left\{(1-\varepsilon)/\varepsilon\right\} L \right\} \geq \zeta_n, L < n/2 \right\},$$

*where $\{\zeta_n\}$ is any arbitrarily diverging sequence. Then for all $n \in \mathbb{N}$ large enough, it holds that*

$$\inf_{\widehat{\eta}} \sup_{P \in \mathcal{P}} \mathbb{E}_P\{d_{\mathrm{H}}(\widehat{\eta}, \, \eta(P))\} \geq \max\left\{ e^{-1}(1-\varepsilon)^{-1}\sigma^2\kappa^{-2}, \, \{2(1-2\varepsilon)e\log((1-\varepsilon)/\varepsilon)\}^{-1} \right\},$$

*where the infimum is over all possible measurable functions of the data and $\eta(P)$ is the set of true change points of $P \in \mathcal{P}$.*

## 2.1 The cost of the contamination

To quantify the cost of the contamination, we compare the results derived above with their counterparts when no contamination presents.

**The difficulty of the problem**. Intuitively speaking, the existence of contamination $H_i$'s increases the difficulty level of detecting the change points in $F_i$'s, and the larger the proportion $\varepsilon$ is the more difficult the problem becomes. Lemma 1 details this cost. When the contamination proportion is small enough to fit in the small $\varepsilon$ regime, no matter how dramatic each contamination distribution $H_i$ is, we are facing a problem with the same difficulty level as if contamination does not exist. When the contamination proportion is large enough, the difficulty is dominated by the difficulty of a one-sample robust mean estimation problem, no matter how large the minimal spacing is.

This is indeed interesting, if not surprising, that the difficulty of this robust change point detection problem has a phase transition between the difficulties of change point without contamination and robust estimation without change points.

**The accuracy of the localisation**. The localisation error will be affected in the presence of adversarial contamination especially when $\varepsilon$ is large relative to $\kappa/\sigma$ as evidenced by Lemma 2. Nevertheless, in terms of order, we should aim to achieve the same localisation accuracy as if no contamination exists.

# 3 The Adversarially Robust Change Point Detection Algorithm

In this section, we propose the adversarially robust change point detection method (ARC), which borrows the strength from robust estimation and standard change point analysis areas.

## 3.1 Methodology

ARC is a very intuitive algorithm but can achieve nearly optimal results in certain regimes as discussed in Section 3.3. We scan through the whole time course using the scan statistic $D_h(\cdot)$, which is the absolute difference between two RUME estimators [22]. The RUME is proposed in the context of one sample robust mean estimation problem based on the idea of shorth estimators [37, 36] and is shown to be simultaneously optimal under both the heavy tailed and Huber $\varepsilon$-contamination model (1). For completeness, we detail the RUME in Algorithm B.1 in the supplementary material. Note that the sample splitting procedure in the RUME helps to avoid statistical dependency during the theoretical analysis but may increase the variance of the estimator.

The scan statistic $D_h(\cdot)$ is a robust variant of the renowned CUSUM statistic [7], except for two differences. First, instead of just using the sample average, with the robustness in mind, a robust mean estimator RUME is deployed. Second, the CUSUM statistic takes the difference between two normalised sample means, which can be of two heavily unbalanced samples. However, due to contamination, it is difficult to track the performance of robust mean estimator based on arbitrary sample sizes. Similar concern also arises in the robust clustering problem where each sub-population needs to be represented sufficiently to derive theoretical guarantees [e.g. 38]. Despite the ubiquity of such scan statistics in the change point literature, arguably, the most closely-related one would be [39], where $\varepsilon$ is set to be zero and $F_i$'s are assumed to be sub-Gaussian.

With $\{D_h(j)\}_j$ in hand, we first focus on all the $4h$-local maximisers – an idea seen in [39] when tackling uncontaminated change point detection problems – defined below. All $4h$-local maximisers are then thresholded by $\lambda > 0$ to avoid overestimating the number of change points.

---

**Algorithm 1** Adversarially robust change point detection (ARC)

**Input:** $\{Y_i\}_{i=1}^n \subset \mathbb{R}$, $\lambda, h > 0$.
  $\mathcal{B} \leftarrow \emptyset, \mathcal{C} \leftarrow \emptyset$;
  **for** $j \in \{2h, 2h+1, \ldots, n-2h\}$ **do**
    $D_h(j) \leftarrow \left| \text{RUME}\left(\{Y_i\}_{i=j+1}^{j+2h}\right) - \text{RUME}\left(\{Y_i\}_{i=j-2h+1}^{j}\right) \right|$;    ▷ *See [22] or Algorithm B.1*
    **if** $j$ is a $4h$-local maximiser of $D_h(j)$ **then**                ▷ *See Definition 1*
      $\mathcal{B} \leftarrow \mathcal{B} \cup \{j\}$;
    **end if**
  **end for**
  **for** $l \in \mathcal{B}$ **do**
    **if** $|D_h(l)| > \lambda$ **then**
      $\mathcal{C} \leftarrow \mathcal{C} \cup \{l\}$;
    **end if**
  **end for**
**Output:** $\mathcal{C}$.

---

**Definition 1.** For any $h \geq 0$ and $x \in \mathbb{R}$, the interval $(x-h, x+h)$ is called the $h$-neighbourhood of $x$. We call $x$ an $h$-local maximiser of a function $f(\cdot)$, if $f(x) \geq f(x')$, for any $x' \in (x-h, x+h)$.

A probably unsatisfactory feature of using the RUME is that the proportion of contamination $\varepsilon$ is required as an input. Unfortunately, this seems to be the bottleneck observed in the majority of optimal robust procedures, including $M$-estimators [e.g. 31], truncated means [40], and more recently developed high-dimensional robust procedures [e.g. 36, 28]. Recently, Chen et al. [35] proposed a tournament-based procedure for one sample robust mean estimation in the Huber $\varepsilon$-contamination model (1) that is adaptive in $\varepsilon$ and it has been used for tuning parameter selection in [33]. We also investigate the empirical performance of our algorithm when using this procedure to select $\varepsilon$ in Section 4. In terms of theory, applying the tournament procedure [35] requires the knowledge of the density function of $F_i$'s and the i.i.d. assumption, which are more restrictive than our framework. Another key tuning parameter in ARC is the window width $h$, the theoretical guidance of which is discussed in Sections 3.2 and 3.3, and the practical guidance can be found in Section 4.

Lastly, we note that the computational complexity of ARC is of order $\mathcal{O}(nh\log(h))$, where the term $h\log(h)$ is from ranking the data to find the shortest interval involved in RUME. Even though the worst case complexity is $\mathcal{O}(n^2\log(n))$, it is still more efficient than the existing methods such as the penalised biweight loss approach with computational complexity $O(n^3)$ (cf. Corollary 2 in [15]).

### 3.2 Theoretical guarantees

**Theorem 1.** *Let $\{Y_i\}_{i=1}^n$ be an independent sequence satisfying Assumption 1 with $\kappa > 0$. Let $\{\widehat{\eta}_k\}_{k=1}^{\widehat{K}}$ be the output of Algorithm 1.*

*Assume that (i) there exists a sufficiently large absolute constant $C_\lambda > 0$ such that $\kappa/\sigma > C_\lambda\sqrt{\max\{\varepsilon, \log(n)/h\}}$; (ii) there exists an absolute constant $C' > 1$ such that $2\varepsilon' + 2\sqrt{C'\varepsilon'\log(n)/h} + C'\log(n)/h < 1/2$, where $\varepsilon' = \max\{\varepsilon, C'\log(n)/h\}$; (iii) the window width satisfies that $h < L/8$; and (iv) the threshold satisfies that $\lambda = C_\lambda\sigma\sqrt{\varepsilon'}$.*

*We then have that there exists an absolute constant $c > 0$ such that*

$$\mathbb{P}\left\{\widehat{K} = K \quad and \quad \max_{k=1}^{\widehat{K}} |\widehat{\eta}_k - \eta_k| \leq 2h\right\} \geq 1 - n^{-c}.$$

Theorem 1 shows that under certain conditions, we have consistent estimation on the number of $f_i$'s change points with $h = o(n)$, in the presence of adversarial attacks. The localisation error rate is essentially the window width $h$ and the required signal-to-noise ratio condition is also a function of the window width $h$. A similar result is obtained in [39] with $h \asymp L$ without contamination. Even though we allow $h = O(L)$ in Theorem 1, this is still somewhat unsatisfactory, but we would like to point out that requiring some knowledge of $L$ is a widely observed phenomenon in the change point analysis literature, even when the contamination is absent. For example, in the wild binary segmentation

algorithm [e.g. 23, 21], random intervals are deployed to localise change points. However, the ideal theoretical performances rely heavily on knowing that the length of these random intervals are of the same order of the minimal spacing $L$ [41].

Given the conditions, we conduct a sanity check on the feasibility of choosing a window width $h$ implying consistency. Based on Proposition D.1 in the supplementary material, we see that the conditions (i)-(iv) in Theorem 1 hold if

$$h > \begin{cases} \max\left\{10,\, 4C_\lambda^2\sigma^2/\kappa^2\right\} C'\log(n), & \varepsilon \le 0.1, \\ w(\varepsilon)C'\log(n), & 0.1 < \varepsilon < 1/4\min\left\{1,\, \kappa^2/(C_\lambda^2\sigma^2)\right\}, \end{cases}$$

where $w(\theta) = 1/(1/2 - \sqrt{2\theta(1-2\theta)})$. The lower bounds on $h$ in both cases are of order $o(n)$. This means that there exist regimes of $h$ such that $n^{-1}\max_k |\widehat{\eta}_k - \eta_k| \to 0$, which implies the consistency.

Theorem 1 provides the guarantees of ARC when there exists at least one change point, i.e. $\kappa > 0$. When $\kappa = 0$, i.e. there is no change point of $f_i$'s, Algorithm 1 is still consistent in the following sense.

**Corollary 1.** *Let $\{Y_i\}_{i=1}^n$ be an independent sequence satisfying Assumption 1 with $\kappa = 0$. Let $\{\widehat{\eta}_k\}_{k=1}^{\widehat{K}}$ be the output of Algorithm 1. Assume that there exists an absolute constant $C' > 1$ such that $2\varepsilon' + 2\sqrt{C'\varepsilon'\log(n)/h} + C'\log(n)/h < 1/2$, where $\varepsilon' = \max\{\varepsilon, C'\log(n)/h\}$; and the thresholding tuning parameter satisfies that $\lambda = C_\lambda\sigma\sqrt{\varepsilon'}$. We then have that there exists an absolute constant $c > 0$ such that $\mathbb{P}\{\widehat{K} = 0\} \ge 1 - n^{-c}$.*

### 3.3 Optimality of the ARC algorithm

Recalling the fundamental limits of the adversarially robust change point detection problem, in view of Theorem 1, ARC can be nearly optimal in terms of both the signal-to-noise ratio condition and the localisation rate in certain regimes, with a properly chosen width $h$ depending on the model parameters. This is indeed restrictive, but generally speaking, robust learning problems suffer from the same restriction. For instance, as we mentioned before, in robust mean estimation problems under the Huber $\varepsilon$-contamination model, the value $\varepsilon$ should be known in order to achieve optimal results in most algorithms and it is impossible to estimate $\varepsilon$ when the contamination distribution is not specified [35]. Padilla et al. [42] studied the nonparametric change point detection problems, which is also a type of robust change point detection problem, and the optimality results thereof rely on the kernel bandwidth $h$ to be the same order as the minimal signal strength. In line with our discussions in Section 2, we consider the following two cases.

**Small $\varepsilon$ regime**. If $\varepsilon \lesssim \log(n)/L$, then with the choice that $h \asymp L$, the conditions in Theorem 1 become $\kappa\sqrt{L}/\sigma \gtrsim \sqrt{\log(n)}$, which by comparing with the small $\varepsilon$ regime in (4) shows that ARC is consistent under the minimal signal-to-noise condition. In this regime, ARC also enjoys the nearly-optimal localisation rate, provided that the minimal segment length satisfies $L \asymp \max\left\{\sigma^2/\kappa^2, 1\right\}\log(n)$.

**Large $\varepsilon$ regime**. Consider two cases in this regime. If $\log(n)/L \lesssim \varepsilon \le 0.1$, then with the choice $h \asymp \log(n)/\varepsilon$, the conditions in Theorem 1 reduce to $\kappa \gtrsim \sigma\sqrt{\varepsilon}$ and $L \gtrsim \log(n)/\varepsilon$, which is the minimal signal-to-noise condition required by comparing to the large $\varepsilon$ regime in (4). In this regime, ARC achieves a nearly-optimal localisation rate of $\log(n)/\varepsilon$ if $\varepsilon$ is of constant order. If $0.1 < \varepsilon < 1/4\min\left\{1, \kappa^2/(C_\lambda^2\sigma^2)\right\}$, then with the choice $h \asymp \omega(\varepsilon)\log(n)$, conditions in Theorem 1 reduce to $\kappa \gtrsim \sigma\sqrt{\varepsilon}$ and $L \gtrsim \omega(\varepsilon)\log(n)$, which is the minimal signal-to-noise condition since $\varepsilon$ is of constant order. In this regime, ARC achieves a nearly-optimal localisation rate of $\omega(\varepsilon)\log(n)$.

## 4 Numerical Results

We consider the following competitors to ARC: PELT, the Pruned Exact Linear Time method [9]; BIWEIGHT, the penalised cost approach based on biweight loss [15]; R_CUSUM, recursive application of a Wald-type testing procedure [43, 15] that is shown to be consistent for single change point detection in a robust regression context; and R_USTAT, a $U$-statistic-type robust bootstrap change point test [24], the theory of which is developed for testing change points but not localisation. PELT serves as a non-robust baseline. As for the other competitors, their theoretical guarantees, if exist,

are established against identical heavy-tail contamination at each time point, but not against the potentially, systematic adversarial attacks.

For BIWEIGHT, we adopt its default setting denoted as BIWEIGHT(2) and a stronger penalty setting BIWEIGHT(5). For R_CUSUM, we combine it with the wild binary segmentation [23], with 500 random intervals and a BIC-type threshold as in [15]. For R_USTAT, we choose the kernel $\nu(x, y) = \text{sign}(x - y)$, the bootstrap sample size 100 and the initial block size 250.

For ARC, the choice of $h$ should depend on the applications. In general, we recommend $h = C \log(n)$ where $10 \leq C \leq 30$. As for simulation purpose, we fix $h = 170$ and $\lambda = \max\{0.6\sigma, 8\sigma\varepsilon\}$. In the simulations, we vary $L$ which serves the purpose of examining the sensitivity of $h$'s choice. More sensitivity results can be found in Section E.2.2 in the supplementary material. Note that the true value of $\sigma$ is used as the input for *all* algorithms in simulation. For real data, due to the suspected short segment length, we consider a range of smaller $h$ and adapt $\lambda$ accordingly to account for the larger estimation error incurred. Different tuning parameter choices for the competitors are also considered. The standard deviations are estimated via the median absolute deviation of the data. For the choice of $\varepsilon$, we use the true $\varepsilon$ as the input in simulations, and a data-driven automated method based on the tournament procedure considered in [35], which views the uncontaminated distributions as Gaussian. We call ARC with automatically-chosen $\varepsilon$ as automated ARC (aARC). Due to the variability of ARC and aARC inherited from the RUME procedure, we run the algorithms over 1000 times on the real data sets and report the mode. See Section E in the supplementary material for further details.

## 4.1 Simulations

Recall that the goal is to detect the changes in the means of $F_i$'s with the attacks in the form of $H_i$'s. We design two settings mimicking two types of adversarial attacks: (i) **creating spurious change points** and (ii) **hiding change points**. Less adversarial settings where the contamination does not rely on the knowledge of change point locations are considered in Section E.2.3 in the supplementary material. Throughout, we let $n = 5000$ and $\delta(\cdot)$ be the Dirac measure.

**(i) Creating spurious change points**. Let $Y_i \sim (1 - \varepsilon)\mathcal{N}(0, \sigma^2) + \varepsilon\delta(-3)$, $i \in \{(j - 1)M + 1, \ldots, (j - 1/2)M\}$, and $Y_i \sim (1 - \varepsilon)\mathcal{N}(0, \sigma^2) + \varepsilon\delta(3)$, $i \in \{(j - 1/2)M + 1, \ldots, jM\}$, $j \in \{1, \ldots, \Delta\}$ and $M = n/\Delta$. An illustration is the left panel in Figure 2. There is no change point on $F_i$'s, but $2\Delta - 1$ changes on $\mathbb{E}[Y_i]$ are created through the contamination distributions $H_i$'s.

**(ii) Hiding change points**. Let $Y_i \sim (1 - \varepsilon)\mathcal{N}(0, 1) + \varepsilon\delta(\kappa/(2\varepsilon))$, $i \in \{(j - 1)M + 1, \ldots, (j - 1/2)M\}$, and $Y_i \overset{i.i.d}{\sim} (1 - \varepsilon)\mathcal{N}(\kappa, 1) + \varepsilon\delta\{\kappa(1 - 1/(2\varepsilon))\}$, $i \in \{(j - 1/2)M + 1, \ldots, jM\}$, $j \in \{1, \ldots, \Delta\}$ and $M = n/\Delta$. An illustration is the right panel in Figure 2, which shows that the change points in $F_i$'s are all offset, in terms of $\mathbb{E}(Y_i)$'s, by the contamination.

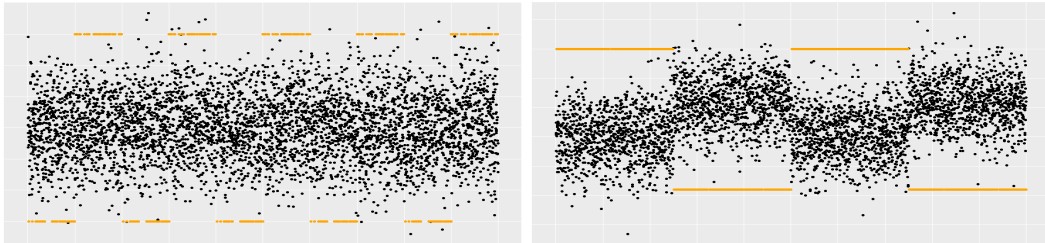

Figure 2: The left panel: an illustration of attack setting (i) with $\varepsilon = 0.1, \Delta = 5$ and $\sigma = 1$. The right panel: an illustration of attack setting (ii) with $\varepsilon = 0.2, \Delta = 2$ and $\kappa = 1.5$. Black dots are realisations of Gaussian distributions and the orange dots are from the contamination distributions.

Note that each scenario in **(i)** is specified by the tuple $\{\varepsilon, \Delta, \sigma\}$ and we consider 19 different combinations. The measurement is $\widehat{K} - K$, with $K = 0$. Each scenario in **(ii)** is specified by the tuple $\{\varepsilon, \Delta, \kappa\}$ and we consider 12 different combinations. The measurements are $|\hat{K} - K|$ and $n^{-1}d_H(\hat{\boldsymbol{\eta}}, \boldsymbol{\eta})$. Detailed results can be found in Section E.2 in the supplementary material. Some representative settings are depicted in Figure 3.

Overall, if the adversarial attacks are creating spurious change points scenario, when the strength of adversarial attack is strong, e.g. when $\varepsilon$ is large and when $\Delta$ and $\sigma$ are small, ARC and aARC

outperform other competitors by detecting fewer spurious change points. If the adversarial attacks are hiding change points, when $\varepsilon$ is large and the signal $\kappa$ is small, the adversarial noise can fool the competitors such that they consistently output incorrect estimated numbers of change points, while ARC and aARC maintain a reasonable performance across all settings.

In terms of localisation errors, in general, we under-perform with respect to the competitors when they can correctly detect the change points, which can be seen as the cost of accuracy when preserving robustness. The observed variability of our results is mainly due to the sample splitting step in the RUME procedure and can be improved with a larger sample size. Moreover, we notice that aARC performs competitively comparing with ARC with the true value $\varepsilon$ as an input.

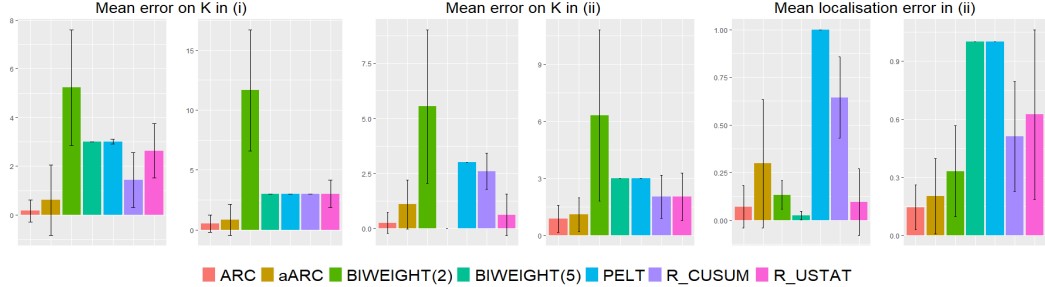

Figure 3: Representative simulation results. From left to right: $|\widehat{K} - K|$ in setting (i) with $\{\varepsilon, \Delta, \sigma\} = \{0.1, 2, 1\}$; $|\widehat{K} - K|$ in setting (i) with $\{\varepsilon, \Delta, \sigma\} = \{0.2, 2, 1\}$; $|\widehat{K} - K|$ in setting (ii) with $\{\varepsilon, \Delta, \kappa\} = \{0.1, 2, 0.66\}$; $|\widehat{K} - K|$ in setting (ii) with $\{\varepsilon, \Delta, \kappa\} = \{0.2, 2, 1.2\}$; $d_H$ in setting (ii) with $\{\varepsilon, \Delta, \kappa\} = \{0.1, 2, 0.66\}$; and $d_H$ in setting (ii) with $\{\varepsilon, \Delta, \kappa\} = \{0.2, 2, 1.2\}$.

## 4.2 Real data analysis

We consider three real data sets in this subsection: the **well-log data set** that has been extensively studied in the existing literature and two **PM2.5 index data sets** that are additions to the literature. Section E.3 in the supplementary material contains additional details on the evaluation method and choices of tuning parameters.

The **well-log data set** [e.g. 44, 15, 45, 46] contains 4049 measurements of nuclear magnetic response during the drilling of a well. We depict the data set and the outputs of aARC in Figure 4. With a few isolated observations, the majority seem to behave well. This falls into the regime of the existing robust change point detection methods. As a result, all competitors considered output very similar results which are omitted in Figure 4.

Consider two **PM2.5 index data sets** [19]: the Beijing PM2.5 index from 15-Apr-2017 to 15-Feb-2021 considered in Section 1 and London PM2.5 index from 1-Jan-2014 to 17-Mar-2021.

In the **Beijing** data set, BIWEIGHT(3), R_CUSUM and aARC all detect two change points based on the **original** data set as shown in the left panel of Figure 1. We then randomly sample 100 points uniformly from the first 1000 data points and change their value to $3\hat{\sigma}$ where $\hat{\sigma}$ is the median absolute deviation of the original data set. Similarly, we sample 50 points from the remaining data and change their value to $0.5\hat{\sigma}$. After this perturbation, only aARC still detects the previous two points and is robust against the spurious one, while the competitors both wrongly miss the previous ones while picking up the spurious one.

The **London** data set, compared to the **well-log data set**, contains way more 'outliers' – in terms of the model (2) – which can be viewed as either that the distributions of interest $F_i$'s are heavy-tailed while $\varepsilon$ is small or $F_i$'s are well-behaved while $\varepsilon$ is large. The result of aARC is similar to that of R_CUSUM, as they treat the data as if the underlying $\varepsilon$ is large. The ARC with an input $\varepsilon = 0.01$ more frequently detects one additional point in the first quarter of the data compared to aARC, which is similar to the result obtained by BIWEIGHT(5), as they treat the data as if the underlying $\varepsilon$ is small.

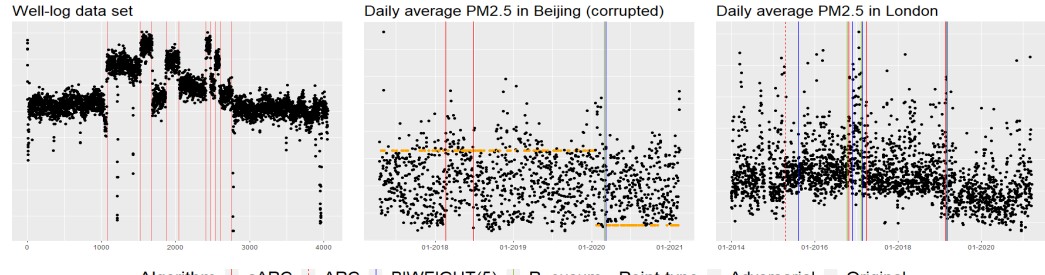

Figure 4: Real data analysis. From left to right: The well-log, Beijing PM2.5 index and London PM2.5 index data sets. Lines are jittered in the middle and right panels for visualisation purposes.

## 5    Conclusion

In this paper, we analysed the change point detection problem under a dynamic extension of the Huber $\varepsilon$-contamination model by allowing the contamination distributions to be different at each point. Under our framework, the adversary can deploy certain attacking strategies that are of interest for the change point detection problem but cannot be modelled within the existing literature. A computationally-efficient algorithm ARC that combines the ideas in robust statistics and change point detection literature is shown to be nearly-optimal in terms of both the signal-to-noise ratio condition and the localisation rate when the minimal spacing $L$ is small or when $\varepsilon$ is of constant order. The optimality results developed in this paper are first time shown in the literature, but are still somewhat restrictive. In order to achieve optimality in the whole parameter space, novel robust estimation and testing techniques are necessary and are on our agenda.

We note that in our framework, even if $\varepsilon$ is of constant order, the optimal localisation error is of the same order compared to the situation when no contamination exists. This is in contrast to the strong contamination model [e.g. 47], where a cluster of outliers are allowed to be created by the adversary, which would unavoidably have a larger impact on the localisation error rate. For example, placing $\varepsilon n$ contaminated points adjacent to a change point would incur a localisation error of $\varepsilon n$ which lead to inconsistent localisation in the sense of (3). Our future plan includes both considering different contamination models and extending the methodology to more challenging data types.

We conjecture that using optimal and efficient robust estimators for high dimensional data or other complex data types combining with the scanning window idea could lead to similar results as in Theorem 1 and Corollary 1. This is left to our future work.

## Supplementary material

The supplementary material contains all the technical details.

## Acknowledgments and Disclosure of Funding

Funding in direct support of this work: DMS-EPSRC EP/V013432/1.

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
