# Adversarially Robust Change Point Detection
# (Supplementary Material)

**Mengchu Li**
Department of Statistics
University of Warwick
mengchu.li@warwick.ac.uk

**Yi Yu**
Department of Statistics
University of Warwick
yi.yu.2@warwick.ac.uk

The supplementary material contains all technical details regarding the proofs of the main results in Sections 2 and 3, further numerical details of Section 4 and additional simulation results.

## A Additional concepts

We use the two-sided Hausdorff distance to measure the distance between the estimated change points and the true change points.

**Definition A.1** (Hausdorff distance). For any subset $S_1$, $S_2 \subset \mathbb{Z}$, the Hausdorff distance $d_H(S_1, S_2)$ between $S_1$ and $S_2$ is defined to be

$$\max \left\{ \max_{s_1 \in S_1} \min_{s_2 \in S_2} |s_1 - s_2|, \max_{s_2 \in S_2} \min_{s_1 \in S_1} |s_1 - s_2| \right\},$$

with the convention that

$$d_H(S_1, S_2) = \begin{cases} \infty, & S_1 = \emptyset \neq S_2 \quad \text{or} \quad S_2 = \emptyset \neq S_1, \\ 0, & S_1 = S_2 = \emptyset. \end{cases}$$

To compare the performance of our method to a range of other methods studied in [1], we consider the following covering metric in Section E.

**Definition A.2** (Covering metric). For any two partitions $\mathcal{G}$ and $\mathcal{G}'$ of the set $\{1, \ldots, n\}$, the covering metric of partition $\mathcal{G}$ by partition $G'$ is defined as

$$C(\mathcal{G}', \mathcal{G}) = \frac{1}{n} \sum_{A \in \mathcal{G}} |A| \max_{A' \in \mathcal{G}'} J(A, A'),$$

where $|A|$ denotes the cardinality of the set $A$ and $J(A, A')$ is the Jaccard index defined as

$$J(A, A') = \frac{|A \cap A'|}{|A \cup A'|}.$$

## B Technical details regarding the Robust Univarite Mean Estimator (RUME)

The RUME is proposed and studied in [2], which is an optimal one-sample robust mean estimator, without contamination. Our ARC algorithm relies on the analysis of RUME, with adaptations to allow for different $H_i$'s at each observation. For completeness, we include all the detailed analysis of RUME in this section, with adaptations to the dynamic Huber contamination model studied in this paper.

Proposition B.1 relies largely on Lemma 3 in [3], except that we consider model (1), which allows the contamination distributions to be different for each $Z_i$. The proof is a minor adaptation from that of Lemma 3 in [3].

35th Conference on Neural Information Processing Systems (NeurIPS 2021).

**Algorithm B.1** Robust Univariate Mean Estimation (RUME)

---

**Input:** $\{Z_i\}_{i=1}^{2h} \subset \mathbb{R}, 0 < \varepsilon < 1, 0 < \delta < 1$

   Randomly split $\{Z_i\}_{i=1}^{2h}$ into $\mathcal{Z}$ and $\mathcal{Z}'$ each containing $h$ points;

$\varepsilon \leftarrow \max\left\{\varepsilon, \dfrac{\log(1/\delta)}{h}\right\}$;

$D \leftarrow \left\lfloor h\left(1 - 2\varepsilon - 2\sqrt{\varepsilon\dfrac{\log(1/\delta)}{h}} - \dfrac{\log(1/\delta)}{h}\right)\right\rfloor$;

$I \leftarrow \emptyset$;

**for** $j \in \{1, \ldots, h - D\}$ **do**

   $I_j \leftarrow Z_{(j+D)} - Z_{(j)}$           $\triangleright$ $Z_{(i)}$ denotes the $i$-th smallest value in $\mathcal{Z}$

   $I \leftarrow I \cup I_j$

**end for**

$j^* \leftarrow$ the index of the smallest value in $I$;

$\hat{I} \leftarrow [Z_{(j^*)}, Z_{(j^*+D)}]$;

$\text{RUME} \leftarrow \dfrac{1}{\sum_{i=1}^{h} \mathbb{1}\{Z_i' \in \hat{I}\}} \sum_{i=1}^{h} Z_i' \mathbb{1}\{Z_i' \in \hat{I}\}$.

**Output:** RUME

---

**Proposition B.1** (Lemma 3 in [3]). *Suppose $Z_1, \ldots, Z_{2h}$ are independent random variables with $Z_i$ generated from the distribution*

$$(1 - \varepsilon_i)F_0 + \varepsilon_i H_i, \quad , i \in \{1, \ldots, 2h\}, \tag{1}$$

*where $\varepsilon_i \leq \varepsilon$, $F_0$ is any distribution in $\mathbb{R}$ with mean $\mu$ and variance upper bounded by $\sigma^2$ and $H_i$'s are any distributions. Let*

$$\varepsilon' = \max\left\{\varepsilon, \frac{\log(1/\delta)}{h}\right\}.$$

*Then, if*

$$2\varepsilon' + 2\sqrt{\varepsilon'\frac{\log(1/\delta)}{h}} + \frac{\log(1/\delta)}{h} < \frac{1}{2} \ \text{ and } \ \delta \leq C'1/h, \tag{2}$$

*where $C' > 0$ is an absolute constant, then it holds that with probability at least $1 - 5\delta$,*

$$|\text{RUME}(\{Z_i\}_{i=1}^{2h}) - \mu| \leq C_1 \sigma\sqrt{\varepsilon'}, \tag{3}$$

*for some absolute positive constant $C_1$.*

*Proof.* Without loss of generality, we can take $\mu = 0$. Let $I^*$ be the interval $(-\sigma/\sqrt{\varepsilon}, \sigma/\sqrt{\varepsilon})$ and $F_0(I^*)$ denotes the probability that one sample drawn from (1) is distributed according to $F_0$ and lies in $I^*$. If $X \sim F_0$, then by Chebyshev's inequality we have

$$\mathbb{P}(|X| > \sigma/\sqrt{\varepsilon}) \leq \varepsilon.$$

Therefore we have $F_0(I^*) = \mathbb{P}(Z_i \in I^* \text{ and } Z_i \sim F_0) = \mathbb{P}(Z_i \in I^* | Z_i \sim F_0)\mathbb{P}(Z_i \sim F_0) \geq (1 - \varepsilon)(1 - \varepsilon) \geq 1 - 2\varepsilon$.

Now let $X_i = \mathbb{1}\{Z_i \sim F_0 \text{ and } Z_i \in I^*\}$ and $F_0^h(I^*) = \sum_{i=1}^{h} X_i/h$. Note that $X_i$ is a Bernoulli random variable with success probability $F_0(I^*)$. Therefore, using the Bernstein inequality for bounded random variables (e.g. Theorem 2.8.4 in [4]), we have with probability at least $1 - \delta$

$$F_0^h(I^*) - F_0(I^*) \geq -\sqrt{F_0(I^*)(1 - F_0(I^*))}\sqrt{\frac{2\log(1/\delta)}{h}} - \frac{2\log(1/\delta)}{3h} \tag{4}$$

$$F_0^h(I^*) \geq 1 - 2\varepsilon - \sqrt{2\varepsilon(1 - 2\varepsilon)\frac{2\log(1/\delta)}{h}} - \frac{2\log(1/\delta)}{3h}, \tag{5}$$

since $F_0(I^*)(1 - F_0(I^*))$ is a decreasing function of $F_0(I^*)$ when $F_0(I^*) > 1/2$. Also, note that the Bernstein bound is used here since it improves the Hoeffding bound when the variance of $X_i$ is small.

Let $g_h(2\varepsilon,\delta) = 2\varepsilon + \sqrt{2\varepsilon(1 - 2\varepsilon)\dfrac{2\log(1/\delta)}{h}} + \dfrac{2\log(1/\delta)}{3h}$ and $\hat{I} = [a,b]$ be the shortest interval containing $h(1 - g_h(2\varepsilon,\delta))$ points in $\mathcal{Z}$. Since $I^*$ also contains at least $h(1 - g_h(2\varepsilon,\delta))$ points due to (5), we must have

$$\text{length}(\hat{I}) \leq \text{length}(I^*) = 2\sigma/\sqrt{\varepsilon}.$$

Further, if $g_h(2\varepsilon,\delta) < 1/2$, then both $\hat{I}$ and $I^*$ contain more than half of the data in $\mathcal{Z}$. As a result, these two intervals must intersect and we have

$$|z - \mu| \leq 4\sigma/\sqrt{\varepsilon} \quad \forall z \in \hat{I}. \tag{6}$$

Next, we control the error of the final estimator. Let $|\hat{I}| = \sum_{Z_i \in \mathcal{Z}'} \mathbb{1}\{Z_i \in \hat{I}\}$ be the number of points from the second sample and lie in $\hat{I}$. Similarly, let $|\hat{I}_H|$ and $|\hat{I}_{F_0}|$ denote the number of points that lie in $\hat{I}$ and are **not** distributed according to $F_0$ (i.e. adversarial point) and according to $F_0$ respectively. Note that

$$\left|\frac{1}{|\hat{I}|}\sum_{Z_i \in \hat{I}} Z_i\right| \leq T_1 + T_2$$

where

$$T_1 = \left|\frac{1}{|\hat{I}|}\sum_{\substack{Z_i \in \hat{I} \\ Z_i \nsim F_0}} Z_i\right| \quad \text{and} \quad T_2 = \left|\frac{1}{|\hat{I}|}\sum_{\substack{Z_i \in \hat{I} \\ Z_i \sim F_0}} Z_i\right|.$$

**Control of $T_1$:** To control $T_1$, we use (6) to get

$$T_1 \leq \frac{|\hat{I}_H|}{|\hat{I}|}\max_{\substack{Z_i \in \hat{I} \\ Z_i \nsim F_0}} |Z_i| \leq \frac{|\hat{I}_H|}{|\hat{I}|}\frac{4\sigma}{\sqrt{\varepsilon}}.$$

To bound the ratio $|\hat{I}_H|/|\hat{I}|$, notice that the total number of points that are drawn from the adversarial distributions can be controlled by the Bernstein inequality. Therefore, we have with probability $1 - \delta$

$$\left|\frac{\hat{I}_H}{h}\right| \leq \varepsilon + \sqrt{\varepsilon(1 - \varepsilon)}\sqrt{\frac{2\log(1/\delta)}{h}} + \frac{2\log(1/\delta)}{3h} = g_h(\varepsilon,\delta), \tag{7}$$

since $|\hat{I}_H|$ is less than the total number of points that are drawn from the adversarial distributions. Together, we have with probability at least $1 - 2\delta$

$$T_1 \leq \frac{g_h(\varepsilon,\delta)}{1 - g_h(2\varepsilon,\delta)}\frac{4\sigma}{\sqrt{\varepsilon}} = C_1\sigma\sqrt{\varepsilon} \tag{8}$$

for some absolute constant $C_1$, where we require $\varepsilon \gtrsim \log(1/\delta)/h$.

**Control of $T_2$:** To control $T_2$, we write it as

$$T_2 = \left|\frac{|\hat{I}_{F_0}|}{|\hat{I}|}\left[\frac{1}{|\hat{I}_{F_0}|}\sum_{\substack{Z_i \in \hat{I} \\ Z_i \sim F_0}} Z_i\right]\right| \leq T_{2a} + T_{2b}$$

where

$$T_{2a} = \frac{|\hat{I}_{F_0}|}{|\hat{I}|}\left[\frac{1}{|\hat{I}_{F_0}|}\sum_{\substack{Z_i \in \hat{I} \\ Z_i \sim F_0}} Z_i - \mathbb{E}[Z|Z \in \hat{I}, Z \sim F_0]\right] \quad \text{and} \quad T_{2b} = \frac{|\hat{I}_{F_0}|}{|\hat{I}|}\left|\mathbb{E}[Z|Z \in \hat{I}, Z \sim F_0]\right|.$$

In $T_{2a}$, since conditional on $Z_i \in \hat{I}$, each $Z_i$ is a bounded random variable with $|Z_i - \mathbb{E}(Z_i)| \leq$ length$(\hat{I}) = 2\sigma/\sqrt{\varepsilon}$ and they are independent of each other, we can again use Bernstein inequality. Using Lemma 15 in [2], which says for any event $E$ that occurs with probability at least $P(E)$,

$$\mathbb{E}_{Z \sim F_0} \left[ (Z - \mathbb{E}[Z|Z \in E])^2 \,|\, Z \in E \right] \leq \frac{\sigma^2}{\mathbb{P}(E)},$$

we can obtain an upper bound for the conditional variance of $Z_i$. Denote $F_0(\hat{I})$ to be the probability that $Z_i$ is distributed according to $F_0$ and lies in $\hat{I}$. Then, we have with probability at least $1 - \delta$

$$T_{2a} \leq \sqrt{\frac{2\sigma^2 \log(2/\delta)}{F_0(\hat{I})|\hat{I}_{F_0}|}} + \frac{4\sigma}{\sqrt{\varepsilon}} \frac{\log(2/\delta)}{3|\hat{I}_{F_0}|}, \tag{9}$$

by the Bernstein inequality.

For $T_{2b}$, we first notice

$$\left| \mathbb{E}_{Z \sim F_0}[Z|Z \notin \hat{I}] \right| = \frac{\mathbb{E}_{Z \sim F_0}[Z \mathbb{1}_{Z \notin \hat{I}}]}{F_0(\hat{I}^c)}$$

$$\leq \frac{\sqrt{\mathbb{E}_{Z \sim F_0}[Z^2] F_0(\hat{I}^c)}}{F_0(\hat{I}^c)}$$

$$= \frac{\sigma}{\sqrt{F_0(\hat{I}^c)}},$$

where $F_0(\hat{I}^c)$ is the probability that $Z$ is distributed according to $F_0$ but does not lie in $\hat{I}$ and we use the Cauchy-Schwarz inequality in the second line. Combining with fact that

$$\left| \mathbb{E}\left[Z|Z \in \hat{I}\right] \right| F_0(\hat{I}) = F_0(\hat{I}^c) \left| \mathbb{E}\left[Z|Z \notin \hat{I}\right] \right|,$$

and assuming $F_0(\hat{I}) \geq 1/2$, we have

$$T_{2b} \leq 2\sigma \sqrt{F_0(\hat{I}^c)}. \tag{10}$$

Combining (8), (9), and (10), we get with probability at least $1 - 3\delta$

$$|\text{RUME} - \mu| \leq C_1 \sigma \sqrt{\varepsilon} + \sqrt{\frac{2\sigma^2 \log(2/\delta)}{F_0(\hat{I})|\hat{I}_{F_0}|}} + \frac{4\sigma}{\sqrt{\varepsilon}} \frac{\log(2/\delta)}{3|\hat{I}_{F_0}|} + 2\sigma \sqrt{F_0(\hat{I}^c)}. \tag{11}$$

To get the claimed bound, we need to study $F_0(\hat{I})$ and $F_0^h(\hat{I}^c) = \dfrac{|\hat{I}_{F_0}|}{\sum_{Z_i \in \mathcal{Z}'} \mathbb{1}_{Z_i \sim F_0}}$. Note that $F_0^h(\hat{I})$ is a sample version of $F_0(\hat{I})$. Let $|\hat{h}_H|$ denotes the number of points in $\mathcal{Z}$ which are **not** drawn from $F_0$ and lie in $\hat{I}$, and $|\hat{h}_{F_0}|$ denotes the number of points in $\mathcal{Z}$ which are drawn from $F_0$ and lie in $\hat{I}$. Note that $|\hat{h}_H|$ and $|\hat{I}_H|$ have the same distribution, therefore using (7), we have with probability at least $1 - \delta$

$$\frac{|\hat{h}_H|}{h} \leq g_h(\varepsilon, \delta).$$

Since $|\hat{h}_H| + |\hat{h}_{F_0}| = h(1 - g_h(2\varepsilon, \delta))$, we have with probability at least $1 - \delta$

$$|\hat{h}_{F_0}| \geq h \left(1 - g_h(2\varepsilon, \delta) - g_h(\varepsilon, \delta)\right). \tag{12}$$

Note that $|\hat{I}_{F_0}|$ and $|\hat{h}_{F_0}|$ also have the same distribution. Therefore, with probability $1 - \delta$,

$$|\hat{I}_{F_0}| \geq h(1 - g_h(2\varepsilon, \delta) - g_h(\varepsilon, \delta)) = C_2 h \tag{13}$$

for some constant $C_2$, where we require $\varepsilon \gtrsim \log(1/\delta)/h$. Equation (12) implies that

$$F_0^h(\hat{I}^c) = \frac{|\hat{I}_{F_0}|}{\sum_{Z_i \in \mathcal{Z}'} \mathbb{1}_{Z_i \sim F_0}} \geq 1 - g_h(2\varepsilon, \delta) - g_h(\varepsilon, \delta).$$

Consequently, we have

$$F_0^h(\hat{I}^c) \leq g_h(2\varepsilon, \delta) + g_h(\varepsilon, \delta) = 3\varepsilon + (\sqrt{\varepsilon(1-\varepsilon)} + \sqrt{2\varepsilon(1-2\varepsilon)})\sqrt{\frac{2\log(1/\delta)}{h}} + \frac{4\log(1/\delta)}{3h}. \tag{14}$$

Provided that $\varepsilon \lesssim \log(1/\delta)/h$, we have

$$F_0^h(\hat{I}^c) \leq 3\varepsilon + C_3 \frac{\log(1/\delta)}{h}. \tag{15}$$

Using the relative deviation lemma from empirical process theory [e.g. Theorem 7 in 5], we can finally bound $F_0(\hat{I}^c)$ as

$$F_0(\hat{I}^c) \leq F_0^h(\hat{I}^c) + 2\sqrt{F_0^h(\hat{I}^c)\frac{\log(S_{\mathcal{F}}(2h)) + \log(4/\delta)}{h}} + 4\frac{\log(S_{\mathcal{F}}(2h)) + \log(4/\delta)}{h}, \tag{16}$$

with probability at least $1 - \delta$. Since the VC dimension for intervals in $\mathbb{R}$ is 2, we have $S_{\mathcal{F}}(2h) \leq (2h+1)^2$ by the Sauer-Shelah Lemma (e.g. Theorem 8.3.16 in [4]). Substituting the upper bound of $S_{\mathcal{F}}(2h)$ and $F_0^h(\hat{I}^c)$ into equation (16) and using the fact that $\sqrt{ab} \leq a + b$ for any $a, b \geq 0$, we get with probability at least $1 - 2\delta$

$$F_0(\hat{I}^c) \leq 9\varepsilon + C_6 \frac{\log(1/\delta)}{h} + 12\frac{\log(2h+1)}{h} = 9\varepsilon + C_6 \frac{\log(1/\delta)}{h} + C_7 \frac{\log(h)}{h}. \tag{17}$$

Combining (11), (13), and (17), we have that with probability at least $1 - 5\delta$

$$|\text{RUME} - \mu| \leq C_1 \sigma\sqrt{\varepsilon} + C_2 \sigma\sqrt{\frac{\log(h)}{h}} + C_3\sqrt{\frac{\log(1/\delta)}{h}} + C_4 \sigma\frac{\log(1/\delta)}{\sqrt{\varepsilon}h}.$$

The claimed bound follows by letting

$$\varepsilon' = \max\left\{\varepsilon, \frac{\log(1/\delta)}{h}\right\}$$

and choosing $\delta \lesssim 1/h$. $\qquad\square$

## C  Proofs of the results in Section 2

In this section, we prove Lemmas 1 and 2. Throughout this section, we use $\delta(x)$ to denote the Dirac measure at point $x$ and consider $\varepsilon_1 = \ldots = \varepsilon_n = \varepsilon$ in Assumption 1. We prove Lemma 1 by considering three sub-problems in Lemmas C.1, C.2 and C.3, respectively. Note that Lemma C.1 is from [6] and it quantifies the difficulty of the change point detection problem without any contamination. The construction in the proof of Lemma C.3 also appears in [7], but we formulate it here formally in Le Cam's framework. Lemma 2 is proved by considering two sub-problems in Lemmas C.4 and C.5.

*Proof of Lemma 1.* Let $s = \max\{8\varepsilon, \log(n)(1-2\varepsilon), 4\varepsilon(1-2\varepsilon)L\}$. To prove Lemma 1, it is sufficient to prove that the claim

$$\inf_{\hat{\eta}} \sup_{P \in \mathcal{P}} \mathbb{E}_P(d_H(\hat{\boldsymbol{\eta}}, \boldsymbol{\eta}(P))) \geq \frac{n}{8} \tag{18}$$

holds in three cases. First, note that when $s = \log(n)(1-2\varepsilon)$, the claim (18) follows from Lemma C.1 below. Next, we show that the claim (18) holds when $s = 8\varepsilon$ in Lemma C.2 below. To conclude the proof, we show that the claim (18) holds when $s = 4\varepsilon(1-2\varepsilon)L$ in Lemma C.3 below. $\qquad\square$

**Lemma C.1** (Lemma 1 in [6]). *Let $\{Y_i\}_{i=1}^n$ satisfy Assumption 1 with $\varepsilon = 0$ and $F_1,\ldots,F_n$ being sub-Gaussian random variables. Let $P_{\kappa,\sigma,L}^n$ denote the corresponding joint distribution. For any $0 < c < 1$, consider the class of distributions*

$$\mathcal{P}_c^n = \left\{ P_{\kappa,\sigma,L}^n : L = \min\left\{ c\frac{\log(n)}{\kappa^2/\sigma^2}, \frac{n}{4} \right\} \right\}.$$

*Then, there exists a $n(c)$, which depends on c, such that, for all n larger than $n(c)$, it holds that*

$$\inf_{\hat{\boldsymbol{\eta}}} \sup_{P \in \mathcal{P}^n} \mathbb{E}_P(d_H(\hat{\boldsymbol{\eta}}, \boldsymbol{\eta}(P))) \geq \frac{n}{8}$$

*where the infimum is over all estimators $\hat{\boldsymbol{\eta}}$ of the change point locations and $\boldsymbol{\eta}(P)$ denotes the change point location of $P \in \mathcal{P}$.*

**Lemma C.2.** *Let $Y_1 \ldots, Y_n$ be a time series satisfying Assumption 1 with only one change point and let $P_{\kappa,L,\sigma,\varepsilon}^n$ denote the corresponding joint distribution. Consider the class of distribution*

$$\mathcal{P} = \left\{ P_{\kappa,L,\sigma,\varepsilon}^n : \frac{\kappa^2 L}{\sigma^2} < \frac{8\varepsilon}{1-2\varepsilon}, L \leq \left\lfloor \frac{n}{4} \right\rfloor \right\},$$

*then*

$$\inf_{\hat{\eta}} \sup_{P \in \mathcal{P}} \mathbb{E}_P(d_H(\hat{\boldsymbol{\eta}}, \boldsymbol{\eta}(P))) \geq \frac{n}{4}$$

*where the infimum is over all estimators $\hat{\eta}$ of the change point locations and $\boldsymbol{\eta}(P)$ is the true change point of $P \in \mathcal{P}$.*

*Proof of Lemma C.2.* Without loss of generality, suppose $n/L = c(n)$, where $c(n)$ is an integer that is allowed to depend on $n$. Denote the density of $\mathcal{N}(u_0, \sigma^2 I)$ by $\phi_0$ and the density of $\mathcal{N}(u_k, \sigma^2 I)$ by $\phi_\kappa$, where $u_0 \in \mathbb{R}^L$ is a vector with all entries being 0, $u_\kappa \in \mathbb{R}^L$ is a vector with all entries being $\kappa$, and $I$ is the identity matrix of dimension $L \times L$. Let $\mathbb{1}_{\phi_\kappa > \phi_0}$ be the indicator function, i.e. $\mathbb{1}_{\phi_\kappa > \phi_0}(x) = 1$ if $\phi_\kappa(x) > \phi_0(x)$ and $\mathbb{1}_{\phi_2 > \phi_1}(x) = 0$ otherwise. Let $\widetilde{P}$ denote the joint distribution of $\{Y_i\}_{i=1}^n$ with one change point at $L \leq n/4$ such that

$$Y_i \sim (1-\varepsilon)F_i + \varepsilon H_i$$

where $F_1 = F_2 = \ldots = F_L = \mathcal{N}(0, \sigma^2)$ and $F_{L+1} = \ldots = F_n = \mathcal{N}(\kappa, \sigma^2)$. For the contamination distributions, we choose the joint distribution of $\{H_i\}_{i=1}^L$ in $\widetilde{P}$ to have the density

$$\frac{1-\varepsilon}{\varepsilon}(\phi_\kappa - \phi_0)\mathbb{1}_{\phi_\kappa > \phi_0},$$

and $\{H_i\}_{i=jL+1}^{(j+1)L}$, in $\widetilde{P}$ to have joint distribution with density

$$\frac{1-\varepsilon}{\varepsilon}(\phi_0 - \phi_\kappa)\mathbb{1}_{\phi_0 > \phi_\kappa},$$

for $j = 1, \ldots, c(n) - 1$.

Similarly, let $\widetilde{Q}$ denote the joint distribution of $\{Y_i'\}_{i=1}^n$ with one change point at $L' = n - L$ such that

$$Y_i \sim (1-\varepsilon)F_i' + \varepsilon H_i'$$

where $F_1' = F_2' = \ldots = F_{L'}' = \mathcal{N}(0, \sigma^2)$ and $F_{L'+1}' = \ldots = F_n' = \mathcal{N}(\kappa, \sigma^2)$. For contamination distributions, we choose $\{H_i'\}_{i=1}^{L'}$ in the same way as $\{H_i\}_{i=L+1}^n$, and $\{H_i'\}_{i=L'+1}^n$ in the same way as $\{H_i\}_{i=1}^L$

Finally, choose the contamination proportion $\varepsilon$ such that

$$\text{TV}\left(\mathcal{N}(u_0, \sigma^2 I), \mathcal{N}(u_\kappa, \sigma^2 I)\right) = \frac{\varepsilon}{1-\varepsilon}.$$

The described joint distributions $\widetilde{P}$ and $\widetilde{Q}$ are indeed belong to $\mathcal{P}$. This can be checked by using the Hellinger distance $H(\cdot, \cdot)$, as a lower bound for the total variation distance

$$
\begin{aligned}
\mathrm{TV}\left(\mathcal{N}(u_0, \sigma^2 I), \mathcal{N}(u_k, \sigma^2 I)\right) &\geq H^2\left(\mathcal{N}(u_0, \sigma^2 I), \mathcal{N}(u_k, \sigma^2 I)\right) \\
&= 1 - \int_{\mathbb{R}^L} \sqrt{\phi_0(x)\phi_k(x)}\, \mathrm{dx} \\
&= 1 - \exp\left(-\frac{1}{8\sigma^2}(u_\kappa - u_0)^T(u_\kappa - u_0)\right) \\
&= 1 - \exp\left(-\frac{1}{8}\frac{\kappa^2 L}{\sigma^2}\right)
\end{aligned}
$$

Therefore, we have

$$
1 - \exp\left(-\frac{1}{8}\frac{\kappa^2 L}{\sigma^2}\right) \leq \frac{\varepsilon}{1-\varepsilon},
$$

which is equivalent to

$$
\frac{\kappa^2 L}{\sigma^2} \leq 8\log\left(1 + \frac{\varepsilon}{1 - 2\varepsilon}\right) \leq \frac{8\varepsilon}{1 - 2\varepsilon}. \tag{19}
$$

Finally, we check that under the construction of $\widetilde{P}$ and $\widetilde{Q}$ we have $\widetilde{Q} = \widetilde{P}$. Notice the identity that

$$
\phi_0 + (\phi_\kappa - \phi_0)\mathbb{1}_{\phi_\kappa > \phi_0} = \phi_\kappa + (\phi_0 - \phi_\kappa)\mathbb{1}_{\phi_0 > \phi_\kappa}. \tag{20}
$$

The joint distribution of $Y_1, \ldots, Y_L$ has density

$$
(1 - \varepsilon)\phi_0 + (1 - \varepsilon)(\phi_\kappa - \phi_0)\mathbb{1}_{\phi_\kappa > \phi_0}, \tag{21}
$$

while the joint distribution of $Y_1', \ldots, Y_L'$ has density

$$
(1 - \varepsilon)\phi_\kappa + (1 - \varepsilon)(\phi_0 - \phi_\kappa)\mathbb{1}_{\phi_0 > \phi_\kappa}. \tag{22}
$$

Therefore, the two joint distributions agree as a result of equation (20). Similarly, the joint distribution of $Y_{jL+1}, \ldots, Y_{(j+1)L}$ has density as in equation (22), for $j = 1, \ldots, c(n) - 1$, and the joint distribution of $Y_{jL+1}', \ldots, Y_{(j+1)L}'$ has density as in equation (21), for $j = 1, \ldots, c(n) - 1$. As a result, we can conclude $\widetilde{Q} = \widetilde{P} \in \mathcal{P}$.

Note that $d_H(\boldsymbol{\eta}(\widetilde{P}), \boldsymbol{\eta}(\widetilde{Q})) \geq n/2$ by construction, therefore by the Le Cam Lemma [e.g. 8], we have

$$
\inf_{\hat{\boldsymbol{\eta}}} \sup_{P \in \mathcal{P}} \mathbb{E}_P(d_H(\hat{\boldsymbol{\eta}}, \boldsymbol{\eta}(P))) \geq \frac{n}{4}\{1 - \mathrm{TV}(\widetilde{P}, \widetilde{Q})\} = \frac{n}{4}.
$$

$\square$

**Lemma C.3.** *Let $Y_1 \ldots, Y_n$ be a time series satisfying Assumption 1 with only one change point and let $P_{\kappa, L, \sigma, \varepsilon}^n$ denote the corresponding joint distribution. Consider the class of distribution*

$$
\mathcal{P} = \left\{P_{\kappa, L, \sigma, \varepsilon}^n : \frac{\kappa}{\sigma} < 2\sqrt{\varepsilon}, L \leq \left\lfloor \frac{n}{4} \right\rfloor\right\},
$$

*then*

$$
\inf_{\hat{\eta}} \sup_{P \in \mathcal{P}} \mathbb{E}_P(d_H(\hat{\boldsymbol{\eta}}, \boldsymbol{\eta}(P))) \geq \frac{n}{4}
$$

*where the infimum is over all estimators $\hat{\boldsymbol{\eta}}$ of the change point locations and $\boldsymbol{\eta}(P)$ is the true change point of $P \in \mathcal{P}$.*

*Proof of Lemma C.3.* Let $\widetilde{P}$ denote the joint distribution of $\{Y_i\}_{i=1}^n$ with one change point at $L \leq n/4$ such that

$$
Y_i \sim (1 - \varepsilon)F_i + \varepsilon H_i
$$

where $F_1 = F_2 = \ldots = F_L = \delta(0)$ and $F_{L+1} = \ldots = F_n$ have the following distribution

$$\begin{cases} F_i = \delta(0) & \text{with probability} \quad 1 - \varepsilon \\ F_i = \delta(\kappa/\varepsilon) & \text{with probability} \quad \varepsilon \end{cases}$$

for $i = L+1, \ldots, n$. Under this setting, we have $f_1 = f_2 = \ldots = f_L = 0$ and $f_{L+1} = \ldots = f_n = \kappa$. For the outlier distributions, we choose $H_1 = \ldots = H_L = \delta(\kappa/\varepsilon)$ while $H_{L+1} = \ldots = H_n$ have the same distribution as $F_n$. Note that by this construction, we have $\{Y_i\}_{i=1}^n$ are independent identically distributed.

Similarly, let $\widetilde{Q}$ denote the joint distribution of $\{Y_i'\}_{i=1}^n$ with one change point at $L' \geq 3n/4$ such that

$$Y_i' \sim (1 - \varepsilon)F_i' + \varepsilon H_i'$$

where we choose $F_1' = \ldots = F_L'$ to be the same as $F_1$ and $F_{L+1}' = \ldots = F_n'$ to be the same as $F_n$. For the outlier distributions, we choose $H_1' = \ldots = H_{L'}'$ to be the same as $H_1$ while $H_{L'+1}' = \ldots, H_n'$ to be the same as $H_n$.

Note that under this construction, we have $\widetilde{P} = \widetilde{Q} \in \mathcal{P}$, since $\{F_i\}_{i=1}^L$ have variance $0$, and $\{F_i\}_{i=L+1}^n$ have variance

$$\kappa^2(1/\varepsilon - 1),$$

which is less equal to $\sigma^2$ under

$$\frac{\kappa}{\sigma} \leq \sqrt{\frac{2\varepsilon}{1 - \varepsilon}} < 2\sqrt{\varepsilon}$$

Since $d_H(\boldsymbol{\eta}(\widetilde{P}), \boldsymbol{\eta}(\widetilde{Q})) \geq n/2$ by construction, therefore by the Le Cam lemma [e.g. 8], we have

$$\inf_{\hat{\boldsymbol{\eta}}} \sup_{P \in \mathcal{P}} \mathbb{E}_P(d_H(\hat{\boldsymbol{\eta}}, \boldsymbol{\eta}(P))) \geq \frac{n}{4}\{1 - \text{TV}(\widetilde{P}, \widetilde{Q})\} = \frac{n}{4}.$$

$\square$

*Proof of Lemma 2.* To prove Lemma 2, we consider two classes of distributions

$$\mathcal{P}_1 = \left\{ P_{\kappa, L, \sigma, \varepsilon}^n : \frac{\kappa^2 L}{\sigma^2} \geq \zeta_n, \ L < \frac{n}{2} \right\},$$

$$\mathcal{P}_2 = \left\{ P_{\kappa, L, \sigma, \varepsilon}^n : (1 - 2\varepsilon) \log\left(\frac{1 - \varepsilon}{\varepsilon}\right) L \geq \zeta_n, \ L < \frac{n}{2} \right\}$$

Notice that $\mathcal{P} = \mathcal{P}_1 \cap \mathcal{P}_2$. Therefore, the proof can be completed in two steps. First, we show in Lemma C.4 below that

$$\inf_{\hat{\boldsymbol{\eta}}} \sup_{P \in \mathcal{P}_1} \mathbb{E}_P(d_H(\hat{\boldsymbol{\eta}}, \boldsymbol{\eta}(P))) \geq \frac{\sigma^2}{\kappa^2} \frac{e^{-1}}{1 - \varepsilon}$$

for all $n$ large enough. Then, we show in Lemma C.5 below that

$$\inf_{\hat{\boldsymbol{\eta}}} \sup_{P \in \mathcal{P}_2} \mathbb{E}_P(d_H(\hat{\boldsymbol{\eta}}, \boldsymbol{\eta}(P))) \geq \frac{1}{2(1 - 2\varepsilon)} \frac{e^{-1}}{\log((1 - \varepsilon)/\varepsilon)}.$$

for all $n$ large enough. Thus, the claim follows. $\square$

**Lemma C.4.** *Let $Y_1 \ldots, Y_n$ be a time series satisfying Assumption 1 with only one change point and let $P_{\kappa, L, \sigma, \varepsilon}^n$ denote the corresponding joint distribution. Consider the class of distribution*

$$\mathcal{P} = \left\{ P_{\kappa, L, \sigma, \varepsilon}^n : \frac{\kappa^2 L}{\sigma^2} \geq \zeta_n, L < \frac{n}{2} \right\},$$

*for any sequence $\{\zeta_n\}$ such that $\lim_{n \to \infty} \zeta_n = \infty$. Then for all $n$ large enough, it holds that*

$$\inf_{\hat{\boldsymbol{\eta}}} \sup_{P \in \mathcal{P}} \mathbb{E}_P(d_H(\hat{\boldsymbol{\eta}}, \boldsymbol{\eta}(P))) \geq \frac{\sigma^2}{\kappa^2} \frac{e^{-1}}{1 - \varepsilon}$$

*where the infimum is over all estimators $\hat{\boldsymbol{\eta}}$ of the change point locations and $\boldsymbol{\eta}(P)$ is the true change point of $P \in \mathcal{P}$.*

*Proof of Lemma C.4.* Let $P_0$ denote the joint distribution of independent random variables $\{Y_i\}_{i=1}^n$ where each $Y_i$ has distribution

$$(1-\varepsilon)F_i^0 + \varepsilon H_i^0.$$

Let

$$F_1^0 = F_2^0 = \ldots = F_L^0 = \mathcal{N}(0, \sigma^2) \quad \text{and} \quad F_{L+1}^0 = F_{L+2}^0 = \ldots = F_n^0 = \mathcal{N}(\kappa, \sigma^2).$$

Similarly, let $P_1$ be the joint distribution of independent random variables $\{Z_i\}_{i=1}^n$ where $Z_i$ has distribution

$$(1-\varepsilon)F_i^1 + \varepsilon H_i^1.$$

Let

$$F_1^1 = F_2^1 = \ldots = F_{L+\Delta}^1 = \mathcal{N}(0, \sigma^2) \quad \text{and} \quad F_{L+\Delta+1}^1 = F_{L+2}^1 = \ldots = F_n^1 = \mathcal{N}(\kappa, \sigma^2),$$

where $\Delta$ is an integer no larger than $n-1-L$. For the adversarial noise distribution, we choose $H_1^0 = \ldots = H_n^0 = H_1^1 = \ldots = H_n^1$, i.e. the contamination distribution is the same across time and is the same for $P_0$ and $P_1$.

By the Le Cam Lemma [e.g. 8] and Lemma 2.6 in [9], we have

$$\inf_{\hat{\eta}} \sup_{P \in \mathcal{P}} \mathbb{E}_P(d_H(\hat{\boldsymbol{\eta}}, \boldsymbol{\eta}(P))) \geq \Delta(1 - \mathrm{TV}(P_0, P_1)) \geq \frac{\Delta}{2} \exp(-\mathrm{KL}(P_0 || P_1)).$$

Since both $P_0$ and $P_1$ are product measures, it holds that

$$\mathrm{KL}(P_0 || P_1) = \sum_{i=L+1}^{L+\Delta} \mathrm{KL}((1-\varepsilon)F_i^0 + \varepsilon H_i^0 || (1-\varepsilon)F_i^1 + \varepsilon H_i^1).$$

Using convexity of the KL divergence (e.g. Lemma 1 in [10]), we have

$$\mathrm{KL}(P_0 || P_1) \leq \sum_{i=L+1}^{L+\Delta} \left((1-\varepsilon)\mathrm{KL}\left(\mathcal{N}(\kappa, \sigma^2) || \mathcal{N}(0, \sigma^2)\right) + \varepsilon \mathrm{KL}\left(H_i^0 || H_i^1\right)\right)$$

$$= \Delta(1-\varepsilon)\frac{\kappa^2}{2\sigma^2} + \varepsilon \sum_{i=L+1}^{L+\Delta} \mathrm{KL}\left(H_i^0 || H_i^1\right)$$

$$= \Delta(1-\varepsilon)\frac{\kappa^2}{2\sigma^2}.$$

since $H_i^0 = H_i^1$, for $i = L+1, \ldots, L+\Delta$. Hence, we have

$$\inf_{\hat{\eta}} \sup_{P \in \mathcal{P}} \mathbb{E}_P(d_H(\hat{\boldsymbol{\eta}}, \boldsymbol{\eta}(P))) \geq \frac{\Delta}{2} \exp\left(-\Delta(1-\varepsilon)\frac{\kappa^2}{2\sigma^2}\right). \tag{23}$$

Next, set $\Delta = \min\{2\sigma^2/(1-\varepsilon)\kappa^2, n-1-L\}$. Using the assumption that

$$\frac{\kappa^2 L}{\sigma^2} \geq \zeta_n.$$

where $\zeta_n$ is a diverging sequence, and

$$\zeta_n > \frac{n/2}{n-1-L} \geq \frac{L}{n-1-L},$$

for all $n$ large enough, we have

$$\frac{(1-\varepsilon)\kappa^2 L}{2\sigma^2} > \frac{\kappa^2 L}{4\sigma^2} > \frac{L}{n-1-L},$$

for all $n$ large enough. Therefore, it must hold that $\Delta = 2\sigma^2/(1-\varepsilon)\kappa^2$ for $n$ large enough and the claimed bound follows from (23). $\qquad \square$

**Lemma C.5.** *Let $Y_1 \ldots, Y_n$ be a time series satisfying Assumption 1 with only one change point and let $P^n_{\kappa, L, \sigma, \varepsilon}$ denote the corresponding joint distribution. Consider the class of distribution*

$$\mathcal{P} = \left\{ P^n_{\kappa, \delta, \sigma, \varepsilon} : (1 - 2\varepsilon) \log\left(\frac{1 - \varepsilon}{\varepsilon}\right) L \geq \zeta_n, L < \frac{n}{2} \right\},$$

*for any sequence $\{\zeta_n\}$ such that $\lim_{n \to \infty} \zeta_n = \infty$. Then for all $n$ large enough, it holds that*

$$\inf_{\hat{\eta}} \sup_{P \in \mathcal{P}} \mathbb{E}_P(d_H(\hat{\boldsymbol{\eta}}, \boldsymbol{\eta}(P))) \geq \frac{1}{2(1 - 2\varepsilon)} \frac{e^{-1}}{\log((1 - \varepsilon)/\varepsilon)}$$

*where the infimum is over all estimators $\hat{\boldsymbol{\eta}}$ of the change point locations and $\boldsymbol{\eta}(P)$ is the true change point of $P \in \mathcal{P}$.*

*Proof of Lemma C.5.* Let $P_0$ denote the joint distribution of $\{Y_i\}_{i=1}^n$ where each $Y_i$ has distribution

$$(1 - \varepsilon)F_i^0 + \varepsilon H_i^0.$$

Let

$$F_1^0 = F_2^0 = \ldots = F_L^0 = \delta(0) \quad \text{and} \quad F_{L+1}^0 = F_{L+2}^0 = \ldots = F_n^0 = \delta(\kappa).$$

The outlier distributions are chosen as

$$H_1^0 = H_2^0 = \ldots = H_L^0 = \delta(\kappa) \quad \text{and} \quad H_{L+1}^0 = H_{L+2}^0 = \ldots = H_n^0 = \delta(0)$$

Similarly, let $P_1$ be the joint distribution of random variables $\{Z_i\}_{i=1}^n$ where each $Z_i$ has distribution

$$(1 - \varepsilon)F_i^1 + \varepsilon H_i^1.$$

Let

$$F_1^1 = F_2^1 = \ldots = F_{L+\Delta}^1 = \delta(0) \quad \text{and} \quad F_{L+\Delta+1}^1 = F_{L+2}^1 = \ldots = F_n^1 = \delta(\kappa),$$

where $\Delta$ is an integer no larger than $n - 1 - L$. The outlier distributions are chosen as

$$H_1^0 = H_2^0 = \ldots = H_{L+\Delta}^0 = \delta(\kappa) \quad \text{and} \quad H_{L+\Delta+1}^0 = H_{L+2}^0 = \ldots = H_n^0 = \delta(0)$$

By the Le Cam Lemma [e.g. 8] and Lemma 2.6 in [9], we have

$$\inf_{\hat{\eta}} \sup_{P \in \mathcal{P}} \mathbb{E}_P(|\hat{\eta} - \eta(P)|) \geq \frac{\Delta}{2}(1 - \mathrm{TV}(P_0, P_1)) \geq \frac{\Delta}{2} \exp(-\mathrm{KL}(P_0, P_1)).$$

Note that, for $i = L + 1, \ldots, L + \Delta$, we have

$$\mathrm{KL}\left((1 - \varepsilon)F_i^0 + \varepsilon H_i^0, (1 - \varepsilon)F_i^1 + \varepsilon H_i^1\right) = (1 - \varepsilon) \log\left(\frac{1 - \varepsilon}{\varepsilon}\right) + \varepsilon \log\left(\frac{\varepsilon}{1 - \varepsilon}\right)$$

$$= (1 - 2\varepsilon) \log\left(\frac{1 - \varepsilon}{\varepsilon}\right).$$

Since both $P_0$ and $P_1$ are product measures, it holds that

$$\mathrm{KL}(P_0 || P_1) = \sum_{i=L+1}^{L+\Delta} \mathrm{KL}((1 - \varepsilon)F_i^0 + \varepsilon H_i^0 || (1 - \varepsilon)F_i^1 + \varepsilon H_i^1).$$

$$= \Delta(1 - 2\varepsilon) \log\left(\frac{1 - \varepsilon}{\varepsilon}\right)$$

Hence, we have

$$\inf_{\hat{\eta}} \sup_{P \in \mathcal{P}} \mathbb{E}_P(d_H(\hat{\boldsymbol{\eta}}, \boldsymbol{\eta}(P))) \geq \frac{\Delta}{2} \exp\left(-\Delta(1 - 2\varepsilon) \log\left(\frac{1 - \varepsilon}{\varepsilon}\right)\right). \tag{24}$$

Next, set $\Delta = \min\left\{ \dfrac{1}{(1 - 2\varepsilon) \log\left(\frac{1-\varepsilon}{\varepsilon}\right)}, n - 1 - L \right\}$. Using our assumption that

$$(1 - 2\varepsilon) \log\left(\frac{1 - \varepsilon}{\varepsilon}\right) L \geq \zeta_n.$$

where $\zeta_n$ is a diverging sequence, and

$$\zeta_n > \frac{n/2}{n-1-L} \geq \frac{L}{n-1-L}.$$

for $n$ large enough, we must have

$$(1-2\varepsilon)\log\left(\frac{1-\varepsilon}{\varepsilon}\right)L > \frac{L}{n-1-L}$$

for $n$ large enough. Therefore, we have $\Delta = \dfrac{1}{(1-2\varepsilon)\log\left(\frac{1-\varepsilon}{\varepsilon}\right)}$ for $n$ large enough, and the claimed bound follows from (24). $\qquad\square$

## D   Proofs of the results in Section 3

In this section, we prove Theorem 1, utilising Proposition B.1 and ideas in [11]. Corollary 1 follows straightforwardly from the proof of Theorem 1. Lastly, Proposition D.1 considers the range of $h$ that satisfies the assumptions in Theorem 1.

*Proof of Theorem 1.* Denote all points that are more than $2h$ away from any true change point by $\mathcal{F}$, i.e. $\mathcal{F} = \{x : |x - \eta_k| > 2h, \forall k = 1, \ldots, K\}$. In the first step, we show that under assumptions (i)-(iii), it holds that for $\forall x \in \mathcal{F}$,

$$\left|\left(\text{RUME}\left(\{Y_i\}_{i=x+1}^{x+2h}\right) - \mathbb{E}[F_x]\right) - \left(\text{RUME}\left(\{Y_i\}_{i=x-2h+1}^{x}\right) - \mathbb{E}[F_x]\right)\right| \leq \lambda \qquad (25)$$

with probability at least $1 - 10\delta$. Note that $x \in \mathcal{F}$ is equivalent to say there is no change point in the interval $[x - 2h, x + 2h]$. Therefore, we have $Y_{x-2h+1}, \ldots, Y_{x+2h}$ are independent random variables with distribution

$$(1 - \varepsilon_i)F_x + \varepsilon_i H_i,$$

for $i = x - 2h + 1, \ldots, x + 2h$. Without loss of generality, we can assume $\eta_k < x < \eta_{k+1}$, then $\mathbb{E}[F_x] = f_{\eta_{k+1}}$. Under the assumption (ii), we can apply Proposition B.1 and a union bound to get for sufficiently large $C_\lambda$ and the choice of $\lambda = C_\lambda \sigma \sqrt{\varepsilon'}$

$$\left|\left(\text{RUME}\left(\{Y_i\}_{i=x+1}^{x+2h}\right) - f_{\eta_{k+1}}\right)\right| \leq \frac{\lambda}{2} \quad \text{and} \quad \left|\left(\text{RUME}\left(\{Y_i\}_{i=x-2h+1}^{x}\right) - f_{\eta_{k+1}}\right)\right| \leq \frac{\lambda}{2}$$

with probability at least $1 - 10\delta$. Consequently, equation (25) follows from the triangle inequality.

Next, we use similar arguments to show that with probability at least $1 - 10\delta$

$$\left|\left(\text{RUME}\left(\{Y_i\}_{i=\eta_k+1}^{\eta_k+2h}\right)\right) - \left(\text{RUME}\left(\{Y_i\}_{i=\eta_k-2h+1}^{\eta_k}\right)\right)\right| > \lambda. \qquad (26)$$

for $\forall k = 1, 2, \ldots, K$. Note that the assumption $L > 8h > 4h$ guarantees that $Y_{\eta_k+1}, \ldots, Y_{\eta_k+2h}$ are independent random variables with distribution $(1 - \varepsilon)F_{\eta_{k+1}} + \varepsilon H_i$ for $i = \eta_k + 1, \ldots, \eta_k + 2h$ where $\mathbb{E}[F_{\eta_{k+1}}] = f_{\eta_{k+1}}$ and $Y_{\eta_k-2h+1}, \ldots, Y_{\eta_k}$ are independent random variables with distribution $(1 - \varepsilon_j)F_{\eta_k} + \varepsilon_j H_j$ for $j = \eta_k - 2h + 1, \ldots, \eta_k$, where $\mathbb{E}[F_{\eta_k}] = f_{\eta_k}$. We take square of the left hand side of equation (26) and rewrite it as

$$\left|\left(\text{RUME}\left(\{Y_i\}_{i=\eta_k+1}^{\eta_k+2h}\right) - f_{\eta_{k+1}}\right) - \left(\text{RUME}\left(\{Y_i\}_{i=\eta_k-2h+1}^{\eta_k}\right) - f_{\eta_k}\right) + (f_{\eta_{k+1}} - f_{\eta_k})\right|^2$$

$$\geq \kappa^2/2 - \left|\left(\text{RUME}\left(\{Y_i\}_{i=\eta_k+1}^{\eta_k+2h}\right) - f_{\eta_{k+1}}\right) - \left(\text{RUME}\left(\{Y_i\}_{i=\eta_k-2h+1}^{\eta_k}\right) - f_{\eta_k}\right)\right|^2,$$

where the inequality follows from the observation that $(x + y)^2 \geq x^2/2 - y^2$ for any $x, y \in \mathbb{R}$. Using Proposition B.1 and triangle inequality, we have

$$\left|\left(\text{RUME}\left(\{Y_i\}_{i=\eta_k+1}^{\eta_k+2h}\right) - f_{\eta_{k+1}}\right) - \left(\text{RUME}\left(\{Y_i\}_{i=\eta_k-2h+1}^{\eta_k}\right) - f_{\eta_k}\right)\right|^2 \leq \lambda^2$$

with probability at least $1 - 10\delta$. Together with the assumption that $\kappa > 2\lambda$, we have

$$\left|\left(\text{RUME}\left(\{Y_i\}_{i=\eta_k+1}^{\eta_k+2h}\right)\right) - \left(\text{RUME}\left(\{Y_i\}_{i=\eta_k-2h+1}^{\eta_k}\right)\right)\right|^2 \geq \kappa^2/2 - \lambda^2 > \lambda^2.$$

In the second step, we consider the following events

$$B_x = \{|D_h(x)| < \lambda\}$$
$$A_{\eta_k} = \{|D_h(\eta_k)| > \lambda\}$$
$$\mathcal{E}_n = \left(\cap_{k=1}^K A_{\eta_k}\right) \cap \left(\cap_{x \in \mathcal{F}} B_x\right)$$

and argue that on the event $\mathcal{E}_n$, we have

$$\hat{K} = K \quad \text{and} \quad \max_{k=1,\ldots,\hat{K}} |\hat{\eta}_k - \eta_k| \leq 2h.$$

Note that it is sufficient to show that on the event $\mathcal{E}_n$ it holds that i) for any estimated change point $\hat{\eta}_k$, $k = 1,2\ldots,\hat{K}$, there is a unique true change point $\eta_k$ lying in the interval $(\hat{\eta}_k - 2h, \hat{\eta}_k + 2h)$ and ii) for each true change point $\eta_k$, $k = 1,2,\ldots,K$, there is a unique estimated change point located in the interval $(\eta_k - 2h, \eta_k + 2h)$.

For i), we notice that $\hat{\eta}_k \in \mathcal{F}^c$ for all $k = 1,2,\ldots,\hat{K}$, according to the definition of event $\mathcal{E}_n$. Therefore, there is at least one true change point in the interval $(\hat{\eta}_k - 2h, \hat{\eta}_k + 2h)$. Using the assumption $L > 8h > 4h$, we see there is at most one true change point in $(\hat{\eta}_k - 2h, \hat{\eta}_k + 2h)$. Therefore i) holds. For ii), using the assumption $L > 8h$, we know every point in the intervals $(\eta_k + 2h, \eta_k + 6h)$ and $(\eta_k - 2h, \eta_k - 6h)$ belong to $\mathcal{F}$. This means $|D_h(x)| < \lambda$ for all x in the aforementioned two intervals. Therefore the $4h$ local maximizers of $|D_h(x)|$ for $x \in (\eta_k - 2h, \eta_k + 2h)$ correspond to the unique local maximizer $\eta^*$ of $|D_h(x)|$ for $x \in (\eta_k - 2h, \eta_k + 2h)$, and we have $|D_h(\eta^*)| \geq |D_h(\eta_k)| > \lambda$.

In the last step, we show that $\mathbb{P}(\mathcal{E}_n^c) \to 0$ using (25) and (26) from step 1. Note that using union bound, we have

$$\mathbb{P}(\mathcal{E}_n^c) \leq \mathbb{P}(\cup_{k=1}^K A_{\eta_k}^c) + \mathbb{P}(\cup_{x \in \mathcal{F}} B_x^c) \leq n \max_k \mathbb{P}(A_{\eta_k}^c) + n \max_{x \in \mathcal{F}} \mathbb{P}(B_x^c). \tag{27}$$

Using (26), we have

$$\max_k \mathbb{P}(A_{\eta_k}^c) = \max_k \mathbb{P}\left(\left|\left(\text{RUME}\left(\{Y_i\}_{i=\eta_k+1}^{\eta_k+2h}\right)\right) - \left(\text{RUME}\left(\{Y_i\}_{i=\eta_k-2h+1}^{\eta_k}\right)\right)\right| < \lambda.\right) \leq 10\delta$$

Using (25), we have

$$\max_{x \in \mathcal{F}} \mathbb{P}(B_x^c) = \mathbb{P}\left(\left|\left(\text{RUME}\left(\{Y_i\}_{i=x+1}^{x+2h}\right) - \mathbb{E}[Y_{x+1}]\right) - \left(\text{RUME}\left(\{Y_i\}_{i=x-2h+1}^{x}\right) - \mathbb{E}[Y_x]\right)\right| > \lambda\right)$$
$$\leq 10\delta.$$

Combining the upper bounds for $\mathbb{P}(A_{\eta_k}^c)$ and $\mathbb{P}(B_x^c)$, we can conclude that

$$\mathbb{P}(\mathcal{E}_n^c) \leq 20n\delta = 20n^{1-C'} \to 0$$

under the choice that $\delta = n^{-C'}$ for some constant $C' > 1$. $\qquad\square$

*Proof of Corollary 1.* Since $\hat{\kappa} = 0$, we have $Y_1,\ldots,Y_n$ are independent random variables with distribution

$$(1 - \varepsilon_i)F_i + \varepsilon_i H_i$$

for $i = 1,\ldots,n$, where $\mathbb{E}[F_i] = f_1$. Using (25) from the Proof of Theorem 1, we have for $x = 2h,\ldots,n - 2h$, it holds that $D_h(x) < \lambda$ with probability at least $1 - 10\delta$. Using the notation from the Proof of Theorem 1, we have on the event $\cap_{x=2h}^{n-2h} B_x$, it holds that $\hat{K} = 0$. It follows from a union bound that

$$\mathbb{P}\left(\bigcup_{x=2h}^{n-2h} B_x^c\right) \leq 10n\delta,$$

Therefore, the claim follows by choosing $\delta = n^{-C'}$ for some constant $C' > 1$. $\qquad\square$

**Proposition D.1.** *Under the same notation as in Theorem 1, the following choices of $h$ can guarantee that assumptions (i)-(iii) holds*

$$h > \max\left\{10,\, 4C_\lambda^2 \frac{\sigma^2}{\kappa^2}\right\} C' \log(n), \qquad if \quad \varepsilon < 0.1,$$

$$h > h(\varepsilon)C' \log(n) \qquad if \quad 0.1 < \varepsilon < \frac{1}{4} \min\left\{1,\, \frac{\kappa^2}{\sigma^2 C_\lambda^2}\right\},$$

*where*

$$h(\theta) = \frac{1}{0.5 - \sqrt{2\theta(1 - 2\theta)}}.$$

*Proof of Proposition D.1.* We consider two separate cases:

1. When $\varepsilon' = \varepsilon$, which is equivalent to

$$h > \frac{C' \log(n)}{\varepsilon},$$

   assumption (ii) can be simplified as

$$2\varepsilon + 2\sqrt{\varepsilon \frac{C' \log(n)}{h}} + \frac{C' \log(n)}{h} < \frac{1}{2},$$

$$\sqrt{\varepsilon} + \sqrt{\frac{C' \log(n)}{h}} < \sqrt{\frac{1}{2} - \varepsilon}$$

$$\text{and} \quad h > \left(\frac{\sqrt{C' \log(n)}}{\sqrt{1/2 - \varepsilon} - \sqrt{\varepsilon}}\right)^2 = h(\varepsilon)C' \log(n).$$

   Note that $h(\varepsilon) \geq 1/\varepsilon$ for $0.1 < \varepsilon < 0.25$. Combining with assumption (i), we have if $0.1 < \varepsilon < \frac{1}{4} \min\left\{1, \frac{\kappa^2}{\sigma^2 C_\lambda^2}\right\}$, then $h > h(\varepsilon) \log(n)$ can satisfy the assumptions for Theorem 1.

2. When $\varepsilon' = C' \log(n)/h$, which is equivalent to

$$h < \frac{C' \log(n)}{\varepsilon},$$

   the assumption (i) requires

$$h > 4C_\lambda^2 C' \frac{\sigma^2}{\kappa^2} \log(n),$$

   and assumption (ii) requires

$$h > 10C' \log(n).$$

   Combining these three conditions, we have if $h$ satisfies

$$\max\left\{10,\, 4C_\lambda^2 \frac{\sigma^2}{\kappa^2}\right\} C' \log(n) < h < \frac{C' \log(n)}{\varepsilon}, \tag{28}$$

   then assumptions in Theorem 1 are satisfied. To ensure that (28) is not an empty set, it is sufficient to require $\varepsilon < 0.1$.

$\square$

# E Further details of the numerical results in Section 4

## E.1 Tuning parameter selection in simulations

For BIWEIGHT [12], we choose (using their notation) the default value as $K = 3\sigma$ and $\beta = 2\sigma^2 \log(n) = 17\sigma^2$, where $\sigma$ is the standard deviation of the uncontaminated distribution $F_i$. We denote this choice of tuning paramter as BIWEIGHT(2). It is noted in [12] that such choice is not guaranteed to ensure consistency. Therefore, we also consider a stronger penalty value $\beta = 5\sigma^2 \log(n) = 42.6\sigma^2$ in the simulation (denoted as BIWEIGHT(5)). For the R_CUSUM procedure, we combine it with the wild binary segmentation framework [13] for multiple change point scenarios. We generate 500 random intervals with threshold set to $5\sigma^2 \log(n) = 42.6\sigma^2$.

For the robust $U$-statistics test, we only consider the univariate version of their proposed statistic $T_n$ here and the distribution of $T_n$ under the null (no change points) is approximated by the Gaussian multiplier bootstrap. Combining with the backward detection algorithm (c.f. Algorithm 1 in [14]), this robust test can be used to perform multiple change point detection.

For aARC, the value $\varepsilon$ is chosen via the tournament procedure proposed by [15]. We include the details here for completeness.

We use the first $T = 300$ of each simulated data set as the training set and obtain a sequence of estimates $\{\hat{\theta}_1, \ldots, \hat{\theta}_m\}$ returned by the RUME with input $\{\varepsilon_1, \ldots, \varepsilon_m\}$ which is an equally spaced set from 0 to 0.25 with size $m = 201$. Consider the following pairwise test function:

$$\phi_{jk} = \mathbb{1}\left\{\left|\frac{1}{T}\sum_{i=1}^{T}\mathbb{1}\left\{p_{\hat{\theta}_j}(Y_i) > p_{\hat{\theta}_k}(Y_i)\right\} - P_{\hat{\theta}_j}\left(Y > \frac{\hat{\theta}_j + \hat{\theta}_k}{2}\right)\right| > \right.$$
$$\left.\left|\frac{1}{T}\sum_{i=1}^{T}\mathbb{1}\left\{p_{\hat{\theta}_j}(Y_i) > p_{\hat{\theta}_k}(Y_i)\right\} - P_{\hat{\theta}_k}\left(Y > \frac{\hat{\theta}_j + \hat{\theta}_k}{2}\right)\right|\right\},$$

where $p_{\hat{\theta}_j}$ is the probability density of $P_{\hat{\theta}_j} = \mathcal{N}(\hat{\theta}_j, \sigma^2)$. When $\phi_{jk} = 1$, then $\hat{\theta}_k$ is favoured over $\hat{\theta}_j$, and when $\phi_{jk} = 0$, then $\hat{\theta}_k$ is favoured over $\hat{\theta}_j$. We select $\varepsilon_{j^*}$ that corresponds to the estimate $\hat{\theta}_{j^*}$ where

$$j^* = \underset{j=1,\ldots,m}{\arg\min} \sum_{k \neq j} \phi_{jk}. \tag{29}$$

It is shown in [15] that the above procedure would pick a $j^*$ such that $P_{\hat{\theta}_{j^*}}$ is close to $\mathcal{N}(\theta, \sigma^2)$ in total variance metric provided that the training data are independent samples from the Huber's $\varepsilon$-contamination model (1) with $F = \mathcal{N}(\theta, \sigma^2)$ and fixed contamination distribution $H$. Note that we consider here specifically the case when $F$, the uncontaminated distribution, is Gaussian in the Huber $\varepsilon$-contamination model (1). Other classes of distributions of $F$ can also be considered by using the corresponding density function.

## E.2 Further simulation results

### E.2.1 Adversarial settings (i) and (ii)

In this subsection, we provide the complete simulation results for the two adversarial attack settings that are considered in our paper. Tables 1 and 2 correspond to scenario (i) and (ii), respectively. The last column in Table 1 shows the mean error in estimating the number of change points and if $\hat{K} = 2\Delta - 1$, it means the algorithm detects the number of spurious change points created by the adversarial noise. The last column in Table 2 shows the median (rescaled) Hausdorff distance and if $\hat{K} = 0$, it means the algorithm cannot detect the true change points in the presence of contamination.

Table 1: Estimated number of change points for various competing methods over 100 simulations when the adversarial noise tries to create spurious change points. The number of change points in terms of $f_i$ is $K = 0$ while the number of change points in terms of $\mathbb{E}[Y_i]$ is $2\Delta - 1$. Bold: methods with the lowest mean error for estimating the number of change point $K$.

| | | | | Number of detected change points | | |
|---|---|---|---|---|---|---|
| $\varepsilon$ | $\Delta$ | $\sigma$ | Method | $\hat{K} = K$ | $\hat{K} = 2\Delta - 1$ | $(\hat{K} - K)/100$ |
| 0 | 1 | 1 | PELT | 100 | 0 | **0.00** |
| | | | ARC | 100 | 0 | **0.00** |
| | | | aARC | 77 | 0 | 0.30 |
| | | | BIWEIGHT(2) | 100 | 0 | **0.00** |
| | | | BIWEIGHT(5) | 100 | 0 | **0.00** |
| | | | R_CUSUM | 100 | 0 | **0.00** |
| | | | R_USTAT | 100 | 0 | **0.00** |
| | | | | | | |
| 0.05 | 1 | 1 | PELT | 0 | 100 | 1.00 |
| | | | ARC | 85 | 15 | 0.15 |
| | | | aARC | 72 | 10 | 0.56 |
| | | | BIWEIGHT(2) | 0 | 72 | 1.50 |
| | | | BIWEIGHT(5) | 1 | 99 | 0.99 |
| | | | R_CUSUM | 97 | 3 | **0.03** |
| | | | R_USTAT | 0 | 88 | 1.15 |
| 0.05 | 1 | 5 | PELT | 95 | 5 | 0.05 |
| | | | ARC | 83 | 16 | 0.18 |
| | | | aARC | 85 | 14 | 0.16 |
| | | | BIWEIGHT(2) | 95 | 5 | 0.05 |
| | | | BIWEIGHT(5) | 100 | 0 | **0.00** |
| | | | R_CUSUM | 100 | 0 | **0.00** |
| | | | R_USTAT | 35 | 55 | 0.79 |
| 0.05 | 1 | 20 | PELT | 100 | 0 | **0.00** |
| | | | ARC | 98 | 2 | 0.02 |
| | | | aARC | 97 | 3 | 0.03 |
| | | | BIWEIGHT(2) | 98 | 2 | 0.02 |
| | | | BIWEIGHT(5) | 100 | 0 | **0.00** |
| | | | R_CUSUM | 100 | 0 | **0.00** |
| | | | R_USTAT | 76 | 15 | 0.34 |
| 0.05 | 5 | 1 | PELT | 0 | 98 | 0.02 |
| | | | ARC | 89 | 0 | 0.15 |
| | | | aARC | 77 | 0 | 0.39 |
| | | | BIWEIGHT(2) | 9 | 4 | 4.75 |
| | | | BIWEIGHT(5) | 96 | 0 | 0.04 |
| | | | R_CUSUM | 100 | 0 | **0.00** |
| | | | R_USTAT | 50 | 6 | 2.14 |
| 0.05 | 5 | 5 | PELT | 100 | 0 | **0.00** |
| | | | ARC | 73 | 0 | 0.36 |
| | | | aARC | 67 | 0 | 0.53 |
| | | | BIWEIGHT(2) | 100 | 0 | **0.00** |
| | | | BIWEIGHT(5) | 100 | 0 | **0.00** |
| | | | R_CUSUM | 100 | 0 | **0.00** |
| | | | R_USTAT | 87 | 0 | 0.24 |
| 0.05 | 5 | 20 | PELT | 100 | 0 | **0.00** |
| | | | ARC | 95 | 0 | 0.05 |
| | | | aARC | 89 | 0 | 0.12 |
| | | | BIWEIGHT(2) | 100 | 0 | **0.00** |
| | | | BIWEIGHT(5) | 100 | 0 | **0.00** |
| | | | R_CUSUM | 100 | 0 | **0.00** |
| | | | R_USTAT | 95 | 0 | 0.12 |
| | | | | | | |
| 0.1 | 1 | 1 | PELT | 0 | 100 | 1.00 |
| | | | ARC | 83 | 12 | **0.24** |
| | | | aARC | 84 | 4 | 0.47 |
| | | | BIWEIGHT(2) | 0 | 37 | 2.96 |
| | | | BIWEIGHT(5) | 0 | 100 | 1.00 |

| | | | | | | |
|---|---|---|---|---|---|---|
| | | | R_CUSUM | 0 | 100 | 1.00 |
| | | | R_USTAT | 0 | 79 | 1.29 |
| 0.1 | 1 | 5 | PELT | 98 | 2 | **0.02** |
| | | | ARC | 72 | 25 | 0.31 |
| | | | aARC | 80 | 20 | 0.20 |
| | | | BIWEIGHT(2) | 27 | 70 | 0.76 |
| | | | BIWEIGHT(5) | 94 | 6 | 0.06 |
| | | | R_CUSUM | 97 | 3 | 0.03 |
| | | | R_USTAT | 0 | 82 | 1.23 |
| 0.1 | 1 | 20 | PELT | 100 | 0 | **0.00** |
| | | | ARC | 96 | 3 | 0.05 |
| | | | aARC | 99 | 1 | 0.01 |
| | | | BIWEIGHT(2) | 100 | 0 | **0.00** |
| | | | Biwerght(5) | 100 | 0 | **0.00** |
| | | | R_CUSUM | 100 | 0 | **0.00** |
| | | | R_USTAT | 51 | 41 | 0.57 |
| 0.1 | 2 | 1 | PELT | 0 | 99 | 3.01 |
| | | | ARC | 86 | 0 | **0.17** |
| | | | aARC | 80 | 3 | 0.61 |
| | | | BIWEIGHT(2) | 0 | 28 | 5.23 |
| | | | Biwerght(5) | 0 | 100 | 3 |
| | | | R_CUSUM | 30 | 20 | 1.43 |
| | | | R_USTAT | 1 | 28 | 2.63 |
| 0.1 | 5 | 1 | PELT | 2 | 80 | 8.07 |
| | | | ARC | 85 | 0 | 0.19 |
| | | | aARC | 85 | 0 | 0.54 |
| | | | BIWEIGHT(2) | 0 | 18 | 11.67 |
| | | | BIWEIGHT(5) | 6 | 10 | 4.42 |
| | | | R_CUSUM | 99 | 0 | **0.02** |
| | | | R_USTAT | 0 | 61 | 9.40 |
| 0.1 | 5 | 5 | PELT | 100 | 0 | **0.00** |
| | | | ARC | 74 | 0 | 0.29 |
| | | | aARC | 52 | 0 | 1.07 |
| | | | BIWEIGHT(2) | 99 | 0 | 0.01 |
| | | | BIWEIGHT(5) | 100 | 0 | **0.00** |
| | | | R_CUSUM | 100 | 0 | **0.00** |
| | | | R_USTAT | 58 | 3 | 1.50 |
| 0.1 | 5 | 20 | PELT | 100 | 0 | **0.00** |
| | | | ARC | 90 | 0 | 0.14 |
| | | | aARC | 97 | 0 | 0.03 |
| | | | BIWEIGHT(2) | 100 | 0 | **0.00** |
| | | | BIWEIGHT(5) | 100 | 0 | **0.00** |
| | | | R_CUSUM | 85 | 0 | 0.29 |
| | | | R_USTAT | 63 | 0 | 0.61 |
| | | | | | | |
| 0.2 | 1 | 1 | PELT | 0 | 100 | 1.00 |
| | | | ARC | 62 | 28 | **0.50** |
| | | | aARC | 56 | 21 | 0.77 |
| | | | BIWEIGHT(2) | 0 | 3 | 9.48 |
| | | | BIWEIGHT(5) | 0 | 100 | 1.00 |
| | | | R_CUSUM | 0 | 100 | 1.00 |
| | | | R_USTAT | 0 | 82 | 1.25 |
| 0.2 | 1 | 5 | PELT | 1 | 99 | 0.99 |
| | | | ARC | 75 | 25 | 0.25 |
| | | | aARC | 99 | 1 | **0.01** |
| | | | BIWEIGHT(2) | 0 | 100 | 1.00 |
| | | | BIWEIGHT(5) | 1 | 99 | 0.99 |
| | | | R_CUSUM | 0 | 100 | 1.00 |
| | | | R_USTAT | 0 | 79 | 1.28 |

| $\varepsilon$ | $\Delta$ | $\kappa$ | Method | $\hat{K}=0$ | $\hat{K}=K$ | median |
|---|---|---|---|---|---|---|
| 0.2 | 1 | 20 | PELT | 100 | 0 | **0.00** |
| | | | ARC | 100 | 0 | **0.00** |
| | | | aARC | 100 | 0 | **0.00** |
| | | | BIWEIGHT(2) | 97 | 3 | 0.03 |
| | | | BIWEIGHT(5) | 100 | 0 | **0.00** |
| | | | R_CUSUM | 100 | 0 | **0.00** |
| | | | R_USTAT | 0 | 80 | 1.27 |
| 0.2 | 2 | 1 | PELT | 0 | 100 | 3.00 |
| | | | ARC | 57 | 2 | **0.53** |
| | | | aARC | 59 | 7 | 0.84 |
| | | | BIWEIGHT(2) | 0 | 3 | 11.64 |
| | | | BIWEIGHT(5) | 0 | 100 | 3.00 |
| | | | R_CUSUM | 0 | 100 | 3.00 |
| | | | R_USTAT | 0 | 33 | 3.01 |
| 0.2 | 5 | 1 | PELT | 0 | 100 | 9.00 |
| | | | ARC | 74 | 0 | **0.36** |
| | | | aARC | 71 | 0 | 0.48 |
| | | | BIWEIGHT(2) | 0 | 5 | 18.48 |
| | | | BIWEIGHT(5) | 0 | 99 | 9.01 |
| | | | R_CUSUM | 0 | 71 | 8.36 |
| | | | R_USTAT | 0 | 70 | 9.52 |
| 0.2 | 5 | 5 | PELT | 100 | 0 | **0.00** |
| | | | ARC | 67 | 0 | 1.56 |
| | | | aARC | 93 | 0 | 0.12 |
| | | | BIWEIGHT(2) | 69 | 0 | 0.54 |
| | | | BIWEIGHT(5) | 100 | 0 | **0.00** |
| | | | R_CUSUM | 100 | 0 | **0.00** |
| | | | R_USTAT | 0 | 51 | 9.26 |
| 0.2 | 5 | 20 | PELT | 100 | 0 | **0.00** |
| | | | ARC | 100 | 0 | **0.00** |
| | | | aARC | 100 | 0 | **0.00** |
| | | | BIWEIGHT(2) | 99 | 0 | 0.01 |
| | | | BIWEIGHT(5) | 100 | 0 | **0.00** |
| | | | R_CUSUM | 100 | 0 | **0.00** |
| | | | R_USTAT | 73 | 1 | 0.99 |

Table 2: Estimated number of change points for various competing methods over 100 simulations when the adversarial noise tries to hide change points. The number of change points in terms of $f_i$ is $K = 2\Delta - 1$ while there is no change points in terms of $\mathbb{E}[Y_i]$. Also the median Hausdorff distance divided by sample size and the number of repetitions that the Hausdorff distance is less than $2h$. Bold: methods with the smallest and second smallest mean error for estimating the number of change point $K$ and median Hausdorff distance divided by sample size.

| $\varepsilon$ | $\Delta$ | $\kappa$ | Method | Number of detected change points | | | $d_H(\hat{\boldsymbol{\eta}}, \boldsymbol{\eta})$ | |
|---|---|---|---|---|---|---|---|---|
| | | | | $\hat{K}=0$ | $\hat{K}=K$ | $|\hat{K}-K|/100$ | $\leq 2h$ | median |
| 0 | 1 | 0.6 | PELT | 0 | 100 | **0.00** | **100** | **0.00** |
| | | | BIWEIGHT(2) | 0 | 100 | **0.00** | **100** | **0.00** |
| | | | BIWEIGHT(5) | 0 | 100 | **0.00** | **100** | **0.00** |
| | | | ARC | 0 | 100 | **0.00** | **100** | **0.00** |
| | | | aARC | 9 | 78 | 0.22 | 75 | **0.01** |
| | | | R_CUSUM | 0 | 98 | **0.00** | **99** | **0.00** |
| | | | R_USTAT | 0 | 82 | 0.30 | 82 | **0.00** |
| 0 | 2 | 0.6 | PELT | 0 | 100 | **0.00** | **100** | **0.00** |
| | | | BIWEIGHT(2) | 0 | 100 | **0.00** | **100** | **0.00** |
| | | | BIWEIGHT(5) | 0 | 100 | **0.00** | **100** | **0.00** |
| | | | ARC | 0 | 100 | **0.00** | **100** | **0.00** |
| | | | aARC | 0 | 88 | 0.12 | **87** | **0.01** |

| | | | Method | | | | | |
|---|---|---|---|---|---|---|---|---|
| | | | R_CUSUM | 0 | 99 | **0.01** | **100** | **0.00** |
| | | | R_USTAT | 0 | 70 | 0.44 | 70 | **0.00** |
| 0.1 | 1 | 0.6 | PELT | 100 | 0 | 1.00 | 0 | 1.00 |
| | | | BIWEIGHT(2) | 39 | 2 | 1.57 | 0 | 0.58 |
| | | | BIWEIGHT(5) | 100 | 0 | 1.00 | 0 | 1.00 |
| | | | ARC | 46 | 36 | 0.70 | 33 | 0.31 |
| | | | aARC | 46 | 32 | 0.96 | 32 | **0.33** |
| | | | R_CUSUM | 60 | 40 | **0.60** | **40** | 0.50 |
| | | | R_USTAT | 0 | 91 | **0.12** | **90** | **0.00** |
| 0.1 | 1 | 0.66 | PELT | 100 | 0 | 1.00 | 0 | 1.00 |
| | | | BIWEIGHT(2) | 0 | 9 | 4.85 | 17 | 0.37 |
| | | | BIWEIGHT(5) | 0 | 100 | **0.00** | **100** | **0.00** |
| | | | ARC | 7 | 93 | **0.07** | **92** | **0.00** |
| | | | aARC | 39 | 52 | 0.59 | 52 | 0.04 |
| | | | R_CUSUM | 8 | 92 | 0.08 | 91 | **0.01** |
| | | | R_USTAT | 0 | 84 | 0.28 | 84 | **0.00** |
| 0.1 | 1 | 1 | PELT | 70 | 2 | 1.05 | 1 | 1.00 |
| | | | BIWEIGHT(2) | 0 | 48 | 1.28 | 52 | **0.03** |
| | | | BIWEIGHT(5) | 0 | 100 | **0.00** | **100** | **0.00** |
| | | | ARC | 3 | 87 | 0.15 | 87 | **0.00** |
| | | | aARC | 2 | 93 | **0.08** | **93** | **0.00** |
| | | | R_CUSUM | 0 | 100 | **0.00** | **100** | **0.00** |
| | | | R_USTAT | 0 | 82 | 0.25 | 82 | **0.00** |
| 0.1 | 2 | 0.6 | PELT | 100 | 0 | 3.00 | 0 | 1.00 |
| | | | BIWEIGHT(2) | 58 | 2 | 2.39 | 0 | 1.00 |
| | | | BIWEIGHT(5) | 100 | 0 | 3.00 | 0 | 1.00 |
| | | | ARC | 35 | 32 | **1.55** | **32** | 0.36 |
| | | | aARC | 29 | 17 | 1.71 | 15 | **0.25** |
| | | | R_CUSUM | 99 | 0 | 2.99 | 0 | 0.75 |
| | | | R_USTAT | 15 | 62 | **0.80** | **54** | **0.05** |
| 0.1 | 2 | 0.66 | PELT | 100 | 0 | 3.00 | 0 | 1.00 |
| | | | BIWEIGHT(2) | 0 | 6 | 5.55 | 21 | 0.12 |
| | | | BIWEIGHT(5) | 0 | 100 | **0.00** | 96 | **0.02** |
| | | | ARC | 0 | 77 | 0.81 | 78 | **0.01** |
| | | | aARC | 14 | 44 | 1.09 | 42 | 0.23 |
| | | | R_CUSUM | 77 | 4 | 2.6 | 4 | 0.75 |
| | | | R_USTAT | 3 | 64 | **0.62** | **59** | 0.05 |
| 0.1 | 2 | 1 | PELT | 73 | 1 | 2.55 | 0 | 1.00 |
| | | | BIWEIGHT(2) | 0 | 57 | 1.47 | 75 | **0.00** |
| | | | BIWEIGHT(5) | 0 | 100 | **0.00** | **100** | **0.00** |
| | | | ARC | 0 | 83 | 0.2 | 83 | **0.01** |
| | | | aARC | 0 | 89 | **0.12** | **89** | **0.01** |
| | | | R_CUSUM | 0 | 100 | **0.00** | **100** | **0.00** |
| | | | R_USTAT | 0 | 70 | 0.43 | 70 | **0.00** |
| 0.2 | 1 | 1.2 | PELT | 100 | 0 | 1.00 | 0 | 1.00 |
| | | | BIWEIGHT(2) | 3 | 0 | 8.01 | 1 | 0.43 |
| | | | BIWEIGHT(5) | 99 | 1 | 0.99 | 0 | 1.00 |
| | | | ARC | 9 | 47 | 0.69 | 42 | 0.13 |
| | | | aARC | 15 | 41 | 1.03 | 33 | 0.24 |
| | | | R_CUSUM | 6 | 94 | **0.06** | **92** | **0.01** |
| | | | R_USTAT | 18 | 68 | **0.38** | **68** | **0.00** |
| 0.2 | 1 | 1.6 | PELT | 94 | 0 | 1.00 | 0 | 1.00 |
| | | | BIWEIGHT(2) | 0 | 0 | 27.61 | 0 | 0.47 |
| | | | BIWEIGHT(5) | 0 | 98 | 0.04 | **98** | **0.00** |
| | | | ARC | 3 | 97 | **0.03** | 97 | **0.01** |
| | | | aARC | 1 | 94 | 0.08 | 94 | **0.01** |
| | | | R_CUSUM | 0 | 100 | **0.00** | **100** | **0.00** |

| | | | | | | | | |
|---|---|---|---|---|---|---|---|---|
| | | | R_USTAT | 0 | 84 | 0.20 | 84 | **0.00** |
| 0.2 | 2 | 1.2 | PELT | 100 | 0 | 3.00 | 0 | 1.00 |
| | | | BIWEIGHT(2) | 7 | 4 | 6.32 | 0 | 0.24 |
| | | | BIWEIGHT(5) | 100 | 0 | 3.00 | 0 | 1.00 |
| | | | ARC | 0 | 29 | **0.88** | **43** | **0.09** |
| | | | aARC | 3 | 29 | **1.09** | **23** | **0.12** |
| | | | R_CUSUM | 52 | 15 | 2.04 | 15 | 0.75 |
| | | | R_USTAT | 56 | 23 | 2.03 | 18 | 1.00 |
| 0.2 | 2 | 1.6 | PELT | 94 | 0 | 2.89 | 0 | 1.00 |
| | | | BIWEIGHT(2) | 0 | 0 | 28.27 | 1 | 0.23 |
| | | | BIWEIGHT(5) | 0 | 98 | 0.06 | **99** | **0.00** |
| | | | ARC | 0 | 98 | **0.02** | 98 | 0.01 |
| | | | aARC | 0 | 95 | 0.05 | 97 | 0.01 |
| | | | R_CUSUM | 0 | 100 | **0.00** | **100** | **0.00** |
| | | | R_USTAT | 2 | 71 | 0.46 | 69 | 0.03 |

### E.2.2 Sensitivity of the choice of $h$

In the simulations above, we use a fixed $h$ for different choices of $\Delta$, which serves the purpose of testing the sensitivity of $h$. To be complete, we also directly test the sensitivity of our methods with respect to the choice $h$ on three settings that are considered in the adversarial setting (i) and (ii) from the previous section. We consider five choices of window width $2h$ from the set $\{10\log(n), 20\log(n), 30\log(n), 40\log(n), 60\log(n)\} = \{85, 170, 255, 340, 511\}$. The results in the tables below are obtained by averaging over 100 repetitions and the numbers in the brackets indicate standard errors. The results suggest that a range of choices of the window width can achieve the best performance, whereas if $h$ is chosen to be too small or large relative to $L$, then the assumptions in our Theorem 1 would be violated and lead to poor performance of the algorithms.

1. Scenario (ii) with $\kappa = 1, \varepsilon = 0.1$ and $\Delta = 2$, which corresponds to $K = 3$ and $L = 1250$.

| Algorithm | Choice of $2h$ | scaled Hausdorff distance | Number of change points |
|---|---|---|---|
| | 85 | 0.11 (0.10) | 3.42 (1.10) |
| | 170 | 0.05 (0.05) | 3.59 (0.89) |
| ARC | 255 | 0.01 (0.02) | 3.05 (0.22) |
| | 340 | 0.01 (0.01) | 3.00(0.00) |
| | 511 | 0.01 (0.00) | 3.00 (0.00) |
| | 85 | 0.12 (0.09) | 4.79 (2.20) |
| | 170 | 0.06 (0.08) | 3.51 (1.19) |
| aARC | 255 | 0.02 (0.05) | 3.03 (0.41) |
| | 340 | 0.02 (0.06) | 2.95 (0.26) |
| | 511 | 0.03 (0.06) | 2.94 (0.24) |

2. Scenario (ii) with $\kappa = 1, \varepsilon = 0.1$ and $\Delta = 3$, which corresponds to $K = 5$ and $L = 833$.

| Algorithm | Choice of $2h$ | scaled Hausdorff distance | Number of change points |
|---|---|---|---|
| | 85 | 0.09 (0.08) | 5.17 (1.21) |
| | 170 | 0.02 (0.03) | 5.32 (0.57) |
| ARC | 255 | 0.01 (0.00) | 5.00 (0.00) |
| | 340 | 0.01 (0.02) | 4.99(0.10) |
| | 511 | 0.17 (0.01) | 3.01 (0.10) |
| | 85 | 0.09 (0.07) | 5.98 (1.96) |
| | 170 | 0.09 (0.04) | 5.27 (0.90) |
| aARC | 255 | 0.03 (0.06) | 4.83 (0.49) |
| | 340 | 0.02 (0.03) | 4.95 (0.21) |
| | 511 | $\infty$ (NaN) | 2.88 (0.41) |

3. Scenario (i) with $\sigma = 1, \varepsilon = 0.1$ and $\Delta = 5$, which corresponds to $K = 0$ and $L = 5000$.

| Algorithm | Choice of $2h$ | Number of change points |
|-----------|----------------|-------------------------|
| ARC | 85 | 6.72 (3.34) |
| | 170 | 2.46 (1.92) |
| | 255 | 0.27 (0.65) |
| | 340 | 0.03(0.22) |
| | 511 | 0.00 (0.00) |
| aARC | 85 | 7.08 (4.29) |
| | 170 | 2.15 (3.53) |
| | 255 | 1.10 (2.11) |
| | 340 | 0.78 (1.74) |
| | 511 | 0.18 (0.76) |

### E.2.3  Less adversarial attack

In below, we further consider two specific attacks which do not use any knowledge about the true change points and therefore less adversarial in nature.

1. We consider the contamination distributions to be the Normal distributions with means $2\sin(10\pi t/T)$ for $t = 1, ..., T$ and $\epsilon = 0.2$, where the sample size T = 3000. Three true change points are equally spaced and located at 750, 1500 and 2250. We fixed the signal-to-noise ratio to be 1.2 with $\kappa = 1.2$ and $\sigma = 1$. Note that the relatively high frequency of the sin function creates the effect of spurious change points on a segment without true change points. We obtain the following results over 100 repetition and the numbers in the brackets indicate standard errors.

| | scaled Hausdorff distance | Number of change points |
|-----------|---------------------------|-------------------------|
| Biweight(2) | 0.16 (0.03) | 9.37 (2.40) |
| Biweight(5) | 0.03 (0.05) | 3.24 (0.52) |
| R_cusum | 0.01 (0.02) | 3.02 (0.14) |
| ARC | 0.05 (0.07) | 2.90 (0.30) |
| aARC | 0.06 (0.09) | 2.88 (0.46) |
| PELT | $\infty$ (NaN) | 0.00 (0.00) |

2. We consider the contamination distributions to be Cauchy distribution with scale parameter 10. The experiment set-up is the same as case 1 above. This heavy-tailed contamination has been considered by [12] and the Biweight algorithm is designed and proven to be effective in this setting. Therefore, it is not surprised that they outperform other methods in the results obtained below.

| | scaled Hausdorff distance | Number of change points |
|-----------|---------------------------|-------------------------|
| Biweight(2) | 0.00 (0.00) | 3.08 (0.27) |
| Biweight(5) | 0.00 (0.00) | 3.00 (0.00) |
| R_cusum | 0.01 (0.00) | 3.03 (0.14) |
| ARC | 0.01 (0.02) | 3.02 (0.14) |
| aARC | 0.01 (0.01) | 3.00 (0.00) |
| PELT | 0.24 (0.01) | 81.74 (12.45) |

Based on the results above, we note that the comparison is unfair for the PELT algorithm as it is not a robust algorithm. The R_cusum algorithm perform competitively but this combination of wild binary segmentation and robust testing procedure has not been studied before (theoretically or empirically). The performances of our algorithms (ARC and aARC) under these less adversarial scenarios are also not bad.

### E.3 Details of Section 4.2

Throughout the real data experiments, we consider choices of $h$ that are smaller than the one used in simulation due suspected short minimal segment lengths and adapt $\lambda = \max\{1.2\sigma\sqrt{5\log(n)h^{-1}}, 8\sigma\varepsilon\}$ to account for the inflated estimation error caused by using a smaller $h$, where $n$ always refers the sample size of the data. Also, we observed that it is not necessary to search for $4h$ local maximisers, which often leads to under estimation of the number of change points. Instead, we search for $2h$ local maximisers in the real data experiments. When implementing aARC, we specify the part of data that we use for selecting $\varepsilon$ according to the strategy described in Section E.1. When implementing ARC, we manually input the value $\varepsilon$ and adapt $\lambda = \max\{1.2\sigma\sqrt{5\log(n)h^{-1}}, 8\sigma\sqrt{\varepsilon}\}$ to account for the larger asymptotic bias when assuming $F_i$'s are heavy-tailed. R_USTAT is not considered in real data analysis since its extension to multiple change point detection is still preliminary and in particular, no method for choosing the initial block size is available.

#### E.3.1 Well-log data

van den Burg and Williams [1] performed a systematic study on the performance of different change point detection algorithms on a selection of real world data sets, which includes the well-log data. Five human annotations were collected for each data set and the covering metric (see Definition A.2) is used to evaluate the distance between the estimated change points and human annotations.

Given the suspected short minimal segment length in the data, we choose $2h = 10\log(n)$ and adjust $\lambda$ accordingly while using the 3500th to 4000th data point to select $\varepsilon$. Running the aARC algorithm 100 times, we achieve an average score of 0.807 under the covering metric, which is better than the result obtained by fine-tuning the BIWEIGHT algorithm as reported in [1]. We note that R_CUSUM implemented with wild binary segmentation also performs competitively achieving a score of 0.849 with 500 intervals and a BIC-type threshold.

#### E.3.2 PM2.5 index data

For both datasets, we choose $2h = 15\log(n)$ and adapt $\lambda$ to account for both the choice of $h$ and the difference between ARC and aARC. We would like to emphasise that the variability of ARC and aARC are mainly due the sample splitting step in the RUME, where we randomly split the $2h$ data points into two sets of size $h$. This step is useful both in terms of theoretical analysis and numerical performance. The variability of the algorithms can be stabilised when the change points possess a stronger signal strength and/or the sample size is large.

Table 3: Frequency of the number of change points detected by aARC and ARC over 1000 repetitions

| Data sets | Method | $\hat{K} = 2$ | $\hat{K} = 3$ | $\hat{K} = 4$ |
|---|---|---|---|---|
| London PM2.5 | ARC | 0 | 350 | 463 |
| | aARC | 0 | 790 | 206 |
| Beijing PM2.5 (corrupted) | aARC | 753 | 126 | 33 |

For the London PM2.5 data, we run aARC and ARC 1000 times and use the 1000th to 1500th data points to select $\varepsilon$ in aARC. The result is shown in Table 3. Out of the 1000 repetitions, the three change points corresponding to the aARC result denoted in Figure 4 are detected 790 times. One additional change point is detected in the first quarter of the data 206 times. While for ARC with input $\varepsilon = 0.01$, four change points, which corresponds to the ARC result denoted in Figure 4, are detected 463 times. BIWEIGHT(2) and BIWEIGHT(3) seem to detect spurious change points caused by the large variability in the data set, as shown in Figure 1.

For the Beijing PM2.5 data set, we run aARC 1000 times on the corrupted data set and use 500th to 1000th data points to select $\varepsilon$. The result is shown in Table 3. The original two change points that were detected on the original data set (without contamination) are detected 753 times on the corrupted the data set and only one of them is detected 86 times. There are 126 times when the spurious change point is detected additionally.

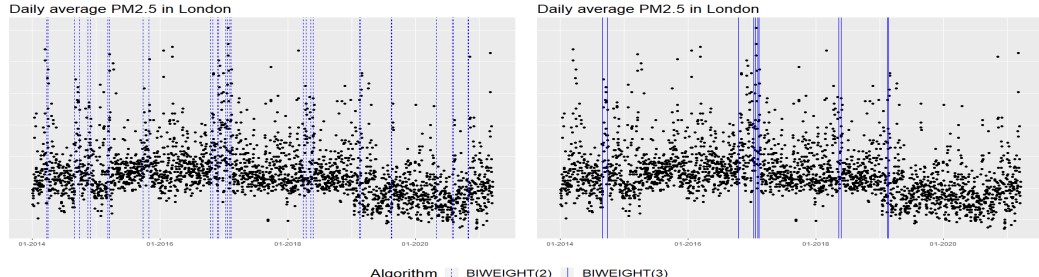

Figure 1: BIWEIGHT method on London PM 2.5 data with different penalty values

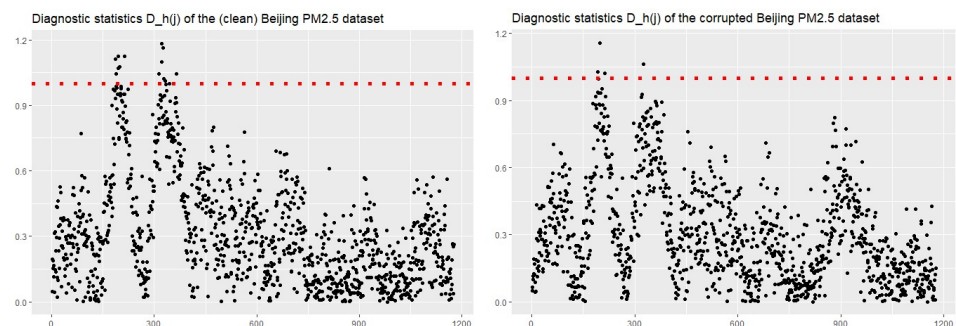

Figure 2: Plot of the diagnostic statistics $D_h(j)$ on the clean and corrupted Beijing PM2.5 index data

An inspection of the plot of the diagnostic statistics on the corrupted and clean data sets in Figure 2 reveals that the original signal is restored in the presence of adversarial attack due to the local nature of the scanning method, while the fake signal created by the adversary will also exceed the threshold occasionally. We also consider the BIWEIGHT method with three different penalty values on the Beijing PM2.5 data. The result is shown in Figure 3. Although different penalty values lead to different segmentation of the original data set, they all detect the spurious change point created by the adversarial noise after the data is contaminated. We use the result of BIWEIGHT(3) for illustration in Figure 1 in Section 1, but we remove the last detected change point for clarity as it seems to be a spurious point near the endpoint of the data set.

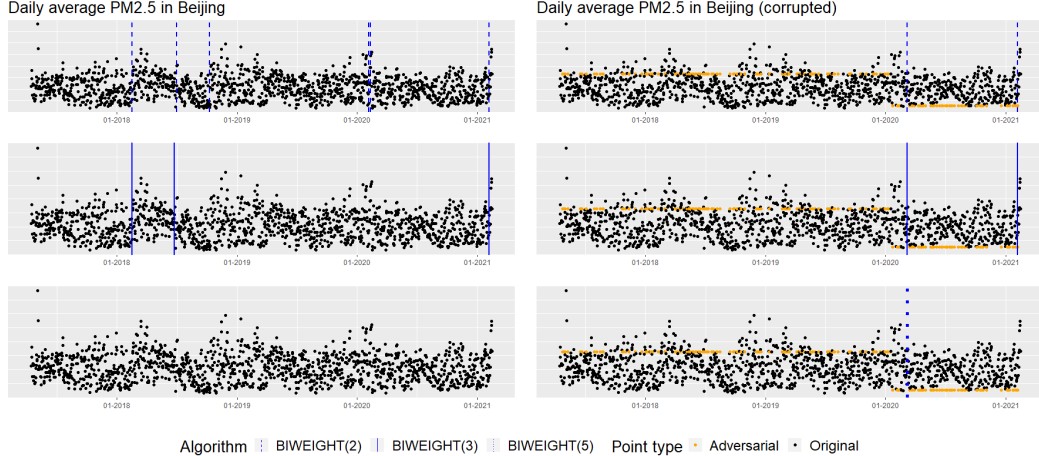

Figure 3: BIWEIGHT method on Beijing PM2.5 data with different penalty values