# OpenReview forum: "Adversarially Robust Change Point Detection"
_NeurIPS.cc/2021/Conference — NeurIPS 2021 Poster_

### Official Review · Reviewer_aT7R · 2021-07-16

**Rating:** 8
**Confidence:** 3

**Summary:**

In this paper, the authors investigate the problem of univariate change point detection under adversarial noise, specifically a dynamic extension of Huber's $\epsilon-$contamination model. The paper also extends the signal-to-noise ratio lower bounds in the non-adversarial setting to an $\epsilon$ dependent bound in the new setting. Also, it provides the optimal localization error rate for the case when the signal-to-noise ratio is above the minimum threshold. Additionally, the authors propose an algorithm that can consistently detect the number of change points when the signal-to-noise ratio is higher than the proposed lower bound, assuming the minimal spacing is at least of the order of $\log(n)$. The provided algorithm uses a mix of RUME and a scanning idea producing an almost optimal localization rate.

**Ethical Concerns:**

I did not find any ethical issues with the paper.

**Limitations And Societal Impact:**

The paper does not seem to have any negative societal impacts and I feel it adequately discusses the incomplete parts of the problem in the conclusion section.

**Main Review:**

Originality: The paper considers a new and relevant setting to study the problem of adversarial univariate change point detection. It also provides novel $\epsilon-$ dependent signal-to-noise ratio lower bounds for detection in this new setting,

Quality: To the best of my knowledge, the submission seems theoretically sound, and the authors do an excellent job of discussing the implications of the theoretical results. The splitting of the problem into two regimes helps better understand the nature of the challenges faced in solving the problem.

Clarity: The paper is very well-written and easy to understand.

Significance: This paper investigates a very relevant problem and gives a great systematic theoretical approach to the problem. Apart from establishing some critical theoretical bounds for the problem, the paper also provides a practically useful algorithm to solve the problem in a near-optimal fashion.

**Time Spent Reviewing:**

4-5

---

> ### Author Response · Authors · 2021-08-10
> **Thank you!**
>
> Thank you very much for your appreciation!

---

### Official Review · Reviewer_k3Ng · 2021-07-16

**Rating:** 4
**Confidence:** 3

**Summary:**

In the paper "Adversarially Robust Change Point Detection", the authors showed that localization rate depends on contamination proportion. In addition, they propose a method for change point detection resistant to such events as, e.g. spurious change points.

**Ethical Concerns:**

-

**Ethics Review Area:**

["I don’t know"]

**Limitations And Societal Impact:**

-

**Main Review:**

In general, although the title and the article's body is full of change point detection and adversarial attacks, it seems both topics are not fully disclosed. We recommend the author concentrate on more interesting topics such as robustness. It helps to bring the work as the development of known methods, e.g. from "Univariate mean change point detection: Penalization, cusum and optimality", from a novel perspective.

In the current variant, the considered cases for adversarial attacks seems contradictory: on the one hand, creating spurious change point is a reasonable attack, but on the other hand, as there is interference in the data distribution by adding the change in it, so why we suspect that methods for change detections shouldn't signalize about this spurious point.

Another limitation is considered the univariate mean change point case. It's an extreme constraint, and we recommend considering multivariate data and changes in scale too.

General comments and questions:
•	What is a motivation for choosing baseline methods, e.g. BIWEIGHT? We recommend using methods from a review of the 2020 year "An Evaluation of Change Point Detection Algorithms" as a benchmark in addition to proposed in the paper.
•	What was a motivation for Hyber $\epsilon$ - contamination model?
•	In line 136, the authors claim the lower bound for $log(n)$. Where it comes from? Please, put some explanations or references.

In general, the text of paper should be improved:
•	Lines and contaminated points are not visible in Figure 1;
•	Line 64. "[22] showed that their … "whom?
•	It will be better not to start sentences with the reference, e.g. lines 74, 76, 77
•	Line 134. $d_H$ defined only in Appendix.
•	Why there are URLs in the references?


**Time Spent Reviewing:**

7

---

> ### Author Response · Authors · 2021-08-10
> **Responses to the comments on the general framework and others.**
>
> Thank you very much for your constructive comments. In the following, we provide responses to the main points that you raised.  We are grateful to you for catching the typos.  Should this paper be accepted, we would incorporate your suggestions in the revision.
>
> __On the general framework__: _[What was a motivation for Huber $\varepsilon$ - contamination model? ...why we suspect that methods for change detections shouldn't signalize about this spurious point?][...we recommend considering multivariate data and changes in scale too.]_
>
> __Answer__: Thank you for your comments. We considered the dynamic extension of the Huber contamination model because of its generality and flexibility in modelling a wide range of adversarial attacks. Given that the existing robust change point detection methods (e.g. BIWEIGHT) are designed against extreme values (i.i.d heavy-tailed errors) and motivated by some surprising adversarial examples in the image classification problem, we are interested in if some structural noise can cause difficulty in the change point detection problem. The dynamic extension of the Huber contamination model offers the flexibility in modelling a range of interesting attacks including the case of creating spurious change points.
>
> As for whether or not the spurious change points created by the adversarial noise should be detected, this is down to the model assumption. In line with the spirit of robustness literature, our task is to detect the existence of the change points in the inliers distributions as described in Assumption 1.
>
> As for the first attempt in such a level of generality, we focus on the univariate case, serving as the blueprint and benchmark for more complex data types, which are our future research directions.
>
>
> __On the motivation of baseline methods__. _[What is a motivation for choosing baseline methods, e.g. biweight? We recommend using methods from a 2020 review paper titled An Evaluation of Change Point Detection Algorithms as a benchmark in addition to proposed in the paper.]_
>
> __Answer__: Thanks for the suggestion. We are indeed aware of the review paper and used their metric to evaluate our algorithm on the well-log data set in Section E.3.1 in the supplementary material.
>
> In our paper, we have four baseline methods, three of which are considered in the review paper.  To be specific, PELT is chosen to serve as a non-robust baseline method; RFPOP is chosen to be a recent robust change point estimation competitor and is referred to as BIWEIGHT in our paper; and WBS, which is a non-robust estimator, is adapted to its robust version and referred to as R\_cusum in our paper.  In addition to these three, we also deploy R\_UStat, which was based on a nonparametric change point testing procedure.  There are other methods discussed in the review paper, we however believe that we have covered a wide range of competitors and therefore refrain from more comparisons.
>
>
> __On the lower bound results__. _[In line 136, the authors claim the lower bound for $\log(n)$. Where it comes from?]_
>
> __Answer__: It follows from Lemma 1 in the paper titled Univariate mean change point detection: Penalization, CUSUM and optimality'.  It is a special case of our Lemma 1 when $\varepsilon = 0$.

---

### Official Review · Reviewer_x4pj · 2021-07-17

**Rating:** 7
**Confidence:** 4

**Summary:**

This paper studies the problem of univariate mean change detection in the presence of adversarial contaminations. The authors formalize the adversarial contamination via the dynamic Huber $\epsilon$-contamination model and propose an algorithm to detect changes. They provide theoretical analysis of the estimation error and perform experiments comparing their method with existing ones.

**Limitations And Societal Impact:**

See comments above for suggestions.

**Main Review:**

The formalization of change point detection with the dynamic Huber $\epsilon$-contamination model is new. The proposed algorithm is a combination of the robust mean estimation and scanning. The theoretical analysis is solid providing two interesting regimes of the detectability as well as near optimal localization error of the proposed algorithm. Experiment results demonstrate the ability of the proposed algorithm against adversarial contaminations. While the paper is mostly clear, I still have several questions/comments stated below.

Major comments:
1. The motivation of considering adversarial contaminations in change point problems is not very convincing. I would like to see more explanation why this setting is interesting. In particular, it seems to me that this contamination is not very realistic -- the attacker needs to have access to the full data sequence, locate potential change points, and then design contaminations. In comparison, the adversarial examples in image classification can simply be obtained by adding Gaussian noise.
2. I found Lemma 2 confusing. In particular, where does the condition (5) come from and why (5) is the complement of the regime in Lemma 1?
3. Following my first comment, since adding specifically designed contaminations is hard, It is interesting see how the proposed method and others perform under less adversarial contaminations.
4. Since $\sigma$ and $\epsilon$ are usually unknown in practice, it is expected to see experiments demonstrating how different the detected change points would be if we estimate these two quantities from data rather than input the true values.

Minor comments:
1. On line 43, "In generally" --> "In general".
2. While the model allows for a different $\epsilon$ at different time point, both Lemma 1 and Lemma 2 assumes $\epsilon_i = \epsilon$. Do they hold true if $\epsilon_i \le \epsilon$?
3. In Algorithm 1, "$j$ is a local maximizer" of what? $D_h(j)$?

**Time Spent Reviewing:**

5

---

> ### Author Response · Authors · 2021-08-10
> **Responses to the comments on motivation, Lemma 2 and others.**
>
> Thank you very much for your appreciation and constructive comments.  In the following, we provide responses to the main points that you raised.  We are grateful to you for catching the typos.  Should this paper be accepted, we would incorporate your suggestions in the revision.
>
> ###
> **Points 1 \& 3**: on motivations. *[...it seems to me that this contamination is not very realistic...In comparison, the adversarial examples in image classification can simply be obtained by adding Gaussian noise.]  [...It is interesting see how the proposed method and others perform under less adversarial contamination.]*
>
> **Answer**:  Thank you for giving us the opportunity to elaborate further on our motivations.  In view of the dynamic Huber contamination model we propose in this paper, we can handle the case that at every time point, a possibly distinct source of contamination is imposed.  This framework includes a wide array of specific adversarial attacks, from less adversarial cases, e.g. simply adding Gaussian noise, to more sophisticated structured contamination, including the case that the adversary may have the knowledge of the change point patterns and locations. As a somewhat practical scenario of the latter, an adversary may plan to exert attack at a true change point location and he/she can add a small amount of carefully designed structural noise to perplex the detection of his/her attack.  The same essence is shared with the popular adversarial examples in image classification where carefully chosen structural noise/perturbation fools the neural networks, e.g. [1].
>
> We also note that under our Assumption 1, the use of the Huber contamination model actually prevents the adversary from seeing the full data sequence. Strictly speaking, he/she must design the contamination prior to the actual data being generated.
>
> [1] ``A Rotation and a Translation Suffice: Fooling CNNs with Simple Transformations " Engstrom et al. 2018.
>
>
> **Point 2**: on Lemma 2. *[I found Lemma 2 confusing. In particular, where does the condition (5) come from and why (5) is the complement of the regime in Lemma 1? ]*
>
> **Answer**: Thank you for your question.  Condition (5) consists of two constraints, corresponding to the signal-to-noise ratio conditions with and without contamination, respectively.  We shall make this more explicit in the revision. In detail, the two terms on the left-hand side correspond to the KL divergence between the two distributions constructed in Lemma C.4 and Lemma C.5 in the supplementary material respectively.
>
> To understand the regimes in the condition (5) and Lemma 1, we note that the lower bound in Lemma 2 is only interesting when $\zeta_n$ is larger than $\log(n)$. Otherwise, the parameters may fall into the regime of Lemma 1 and in this case, the lower bound in both Lemma 1 and Lemma 2 holds. When (5) holds and $\zeta_n$ is larger than $\log(n)$, we have indeed the parameters fall in the complement of the regime of Lemma 1. Having said this, we do acknowledge that condition (5) is not the exact complement of the regime specified in Lemma 1, due to the presence of both change points and contamination.
>
>
> **Point 4**: on the estimation of $\sigma$ and $\varepsilon$. *[... how different the detected change points would be if we estimate $\sigma$ and $\varepsilon$ from data rather than input the true values.]*
>
> **Answer**:  Thank you for your question.  The median absolute deviation has been widely used in the change point literature to estimate $\sigma$ and we also used it for real data sets (line 288).  We proposed the automatic ARC (aARC) algorithm in the paper, which is indeed designed for estimating $\varepsilon$ from the data set. More details can be found in Section E.1 in the supplementary material.  Both simulated data sets and real data sets have shown that our aARC performs competitively.
>
>
> **Minor point 2** *[Lemma 1 and Lemma 2 assumes $\varepsilon_i = \varepsilon$. Do they hold true if $\varepsilon_i \leq \varepsilon$?]*
>
> **Answer**: Due to the artefact of our current proof, unfortunately, $\varepsilon_i = \varepsilon$ is required.  To be specific, the requirement that $\varepsilon_i \geq \varepsilon$ is only involved in a logarithmic term.  We will try to improve the proofs to avoid this constraint and only require that $\varepsilon_i \leq \varepsilon$ in the revision.
>
>
> **Minor point 3** *[``$j$ is a local maximizer of what? $D_h(j)$?"]*
>
> **Answer**: Thanks for catching this. In Algorithm 1, $j$ is a local maximizer of $D_h(j)$.

---

> > ### Comment · Reviewer_x4pj · 2021-08-21
> > **Follow-Up Question**
> >
> > Thank you for your response. I have one follow-up question.
> >
> > Regarding my comments 1 & 3, what concerns me is that, in your motivation and experiments, the attackers are assumed to know the true change point. How can the attackers know the true change point if they don't have access to the full data? Do you have any practical examples that lie in this setting? If this type of attacks is less common, then, in my opinion, adding experiments comparing the performance under less adversarial contaminations (i.e., the attacker does not know the true change point) would be a good addition to the paper.

---

> > > ### Author Response · Authors · 2021-08-24
> > > **Response to the follow-up question**
> > >
> > > Thank you for your follow-up questions.
> > >
> > > **Point 1**: On the adversary knowledge *[How can the attackers know the true change point if they don't have access to the full data?]*
> > >
> > > **Answer**: Assumption 1 describes the generative model considered in our paper, which includes the locations of the change points.  An adversary is implicitly assumed to exist and has the full knowledge of the generative model and therefore he/she knows the change point locations before the data being generated. Although it is hard to assess its practicality, this type of adversary (having the knowledge of the underlying model) is common in the robust statistics literature. For example in robust mean estimation [e.g. 1] and robust testing problems [e.g. 2], the knowledge of underlying models is used to construct worst-case contamination which then motivates optimal estimation procedures.
> > >
> > > **Point 2**: On less adversarial contamination settings (those do not involve the knowledge of true change points).
> > >
> > > **Answer**: We agree that there is no “one-size-fits-all” notion of adversary and it is likely that the problem at hand is more suitable to be modelled by some 'less adversarial contamination' setting. Due to the flexibility of our method, there indeed exist situations where we cannot outperform those methods which are designed specifically. In this response (and also in the paper), we aim to convey the message that our methods are robust across a wide range of settings.
> > >
> > > To be specific, in the paper, the attack of creating spurious change points does not *necessarily* require the knowledge of specific locations of change points.  In this response, we add two additional settings, in which the attack strategies do not use any knowledge about the true change points and we will include them in the final version of the paper.
> > >
> > > **Setting 1**  We consider the contamination distributions to be the Gaussian distributions with means $2\sin(10\pi t/T)$ and variance $1$, for $t = 1, \ldots, T$ and $\epsilon = 0.2$, where the sample size $T = 3000$. Three true change points are equally spaced and located at $750$, $1500$ and $2250$.  The inliers' distributions are Gaussian distributions with variance $1$. We fix the signal-to-noise ratio to be $1.2$.  The relatively high frequency of the trigonometric function creates the effect of spurious change points on segments without true change points. We obtain the following results over 100 repetitions and the numbers in the brackets indicate standard errors.
> > >
> > > |             | Scaled Hausdorff distance | Number of change points |
> > > |-------------|---------------------------|-------------------------|
> > > | Biweight(2) | 0.16 (0.03)               | 9.37 (2.40)             |
> > > | Biweight(5) | 0.03 (0.05)               | 3.24 (0.52)             |
> > > | R\_cusum    | 0.01 (0.02)               | 3.02 (0.14)             |
> > > | ARC         | 0.05 (0.07)               | 2.90 (0.30)             |
> > > | aARC        | 0.06 (0.09)               | 2.88 (0.46)             |
> > > | PELT        | $\infty$ (NaN)            | 0.00 (0.00)             |
> > >
> > >  **Setting 2**  We consider the contamination distributions to be Cauchy distribution with scale parameter $10$. The rest of the experiment setup is the same as the case above. This heavy-tailed type contamination has been considered by [3] and the Biweight algorithm is designed specifically for this setting.
> > >
> > > |             | Scaled Hausdorff distance | Number of change points |
> > > |-------------|---------------------------|-------------------------|
> > > | Biweight(2) | 0.00 (0.00)               | 3.08 (0.27)             |
> > > | Biweight(5) | 0.00 (0.00)               | 3.00 (0.00)             |
> > > | R\_cusum    | 0.01 (0.00)               | 3.03 (0.14)             |
> > > | ARC         | 0.01 (0.02)               | 3.02 (0.14)             |
> > > | aARC        | 0.01 (0.01)               | 3.00 (0.00)             |
> > > | PELT        | 0.24 (0.01)               | 81.74 (12.45)           |
> > >
> > >
> > > Based on the above additional results, we can see that in a less contaminated setting (**Setting 1**), both of our proposed methods (ARC and aARC) outperform Biweight(2) and Biweight(5), which overestimate the number of change points.  R\_cusum performs the best, but this combination of wild binary segmentation and robust testing procedure has not been studied before (theoretically or empirically).  In the least contaminated setting  (**Setting 2**), we introduce i.i.d. heavy-tailed error, which is the setting specifically targeted by Biweight.  It is expected that they perform the best, but our methods still perform reasonably. Finally, we would like to point out that the comparison is unfair for the PELT algorithm as it is not a robust algorithm.
> > >
> > >
> > > [1] 'Robust Estimators in High Dimensions without the Computational Intractability. ' Diakonikolas et al., 2019.
> > >
> > > [2] 'Robust confidence limits.' Huber, 1968.
> > >
> > > [3] 'Changepoint Detection in the Presence of Outliers.' Fearnhead and Rigaill, 2019.

---

> > > > ### Comment · Reviewer_x4pj · 2021-08-25
> > > > **Thank you for your response**
> > > >
> > > > Thank you for your response. It is nice to see that the proposed method performs reasonable well under less adversarial contaminations. I am happy with the response and will increase my score.

---

### Official Review · Reviewer_9vuk · 2021-07-28

**Rating:** 7
**Confidence:** 3

**Summary:**

This paper studies the change point detection problem in the context of possibly adversarial contamination. Recall that previous literature on change point detection either neglected the robustness issues, or focused on heavy-tailed data only. Under a generalization of the Huber contamination model (Assumption 1) authors first derive lower bounds (Lemmas 1 and 2) which highlight an interesting transition phase between the impacts of the signal to noise ratio and the proportion of outliers. Next an algorithm is proposed, mixing standard change point detection method with a robust mean estimator (Algorithm 1). Consistency guarantees for Algorithm 1 are derived in Theorem 1. Finally, numerical experiments are presented, showing the robustness of the proposed method to adversarial contamination (Section 4).

**Limitations And Societal Impact:**

Yes

**Main Review:**

I find the paper globally well written and interesting. The combination of change point detection and outlier robustness is quite natural, and the solution proposed is very sound. The phase transition phenomenon exhibited is interesting, and well discussed. The fact that no knowledge about $\epsilon$ is required is of great practical interest. On the negative side:
- l. 50-52: works on change point detection in the context of (time independent) adversarial contamination are evoked, but their results are not compared to Theorem 1 for instance. Furthermore, from the reading of Section 1.1, I kind of understood that this paper was the first one to focus on change point detection and robustness to outliers (independently of the dependence on $i$). Could the authors be a bit more precise on this point?
- I would like to point out the following work [3], as the $O \cup I$ framework developed therein seems very similar to Assumption 1. In particular, no i.i.d. assumption is made for the outliers.
- In the same vein, I wanted to point out the Median-of-Means (MoM), a robust mean estimator, firstly shown to be robust to heavy-tailed data [1, 2], and now studied under adversarial contamination too [3, 4]. This could be of interest for the authors and future research
- Has the transition phase phenomenon been also verified empirically?
- The interpretation of Theorem 1 is made slightly difficult by the fact that most constants are not explicited. This is true in particular for the value of $c$, which controls the rate. Is there any analogous result for time independent contamination, that could serve as bechmark?
- l. 240: I might have missed something, but $\kappa = 0$ only means that the smallest gap is 0, not that there is no change point, no?

Minor comments:
- Figures and Algorithms on top/bottom of pages ease the reading
- l. 43: In general, three
- l. 186: of *the* estimator
- l. 226: wi*n*dow
- l. 227: without contamination

Overall, I feel the paper develops a coherent story with an interesting solution, motivating my grade. However, I am not extremely familiar with the related works so I keep a 3 confidence and wait for discussion with other reviewers to refine my assessment.

[1] "Sub-gaussian mean estimators" Devroye et al. 2016

[2] "Loss minimization and parameter estimation with heavy tails" Hsu and Sabato 2016

[3] "Robust classification via MOM minimization" Lecué et al.2020

[4] "Generalization Bounds in the Presence of Outliers: a Median-of-Means Study" Laforgue et al. 2021

**Post-rebuttal edit**

I thank the authors for their response. I encourage them to incorporate some of the arguments raised (especially for points 1&5 and points 2&3) in the revision, as things were more clear with these explanations. In light of the other reviews (and the subsequent discussions with the authors) I keep my 7 score and am in favor of acceptance.

**Time Spent Reviewing:**

5

---

> ### Author Response · Authors · 2021-08-10
> **Responses to the comments on Theorem 1, alternative robust estimators and others.**
>
> Thank you very much for your appreciation and constructive comments.  In the following, we provide responses to the main points that you raised.  We are grateful to you for catching the typos.  Should this paper be accepted, we would incorporate your suggestions in the revision.
>
>
> __Points 1 \& 5__: on Theorem 1. _[...most constants are not explicit in Theorem 1 ... Is there any analogous result for time independent contamination, that could serve as benchmark?] [... Is this paper the first one to focus on change point detection and robustness to outliers (independently of the dependence on $i$)? ]_
>
> __Answer__ : Thank you for your question. We shall make this more explicit in the revision, should this paper be accepted.  We are indeed the first (to the best of our knowledge) to formalise the contamination through a (dynamic) Huber $\varepsilon$-contamination framework, while the previous work (e.g. Fearnhead and Rigaill, 2019, Chen and Yu, 2019) mainly focuses on additive heavy-tailed noise. In particular, we model the contamination distribution to be different at each point which allows the adversary to design structural attacks (e.g. creating spurious change point).
>
> Regarding the comparison of Theorem 1 with the existing literature, we note that if we degenerate our result to the case $\varepsilon = 0$ (no contamination) and the case $\kappa = 0$ and same contamination (no change point), then Theorem 1 matches those in the standard change point and standard robustness literature, respectively.  However, we are not able to find existing literature with fixed sample results for Huber contamination (same $\varepsilon$) change point models.
>
> __Points 2 \& 3__: on other literature. _[... the $O\cup I$ framework developed therein seems very similar to Assumption 1. In particular, no i.i.d. assumption is made for the outliers.] [...the Median-of-Means (MoM) ... could be of interest for the authors and future research.]_
>
> __Answer__:  Thank you for bringing these references to us and we are indeed grateful for the suggestions of future research directions including using the MoM estimators. Regarding the popular $O\cup I$ framework, we referred it as the strong contamination model in lines 359-360 and as we briefly pointed out in the last paragraph of the conclusion section, the $O\cup I$ framework and the dynamic Huber framework (Assumption 1) are quite different for localising change points.
>
> Intuitively, Assumption 1 requires the contamination to `spread evenly' across the data sequence while the $O\cup I$ framework allows the adversary to manipulate a consecutive sequence of points.  To be specific, in the $O\cup I$ framework, $\varepsilon n$ contaminated points can be placed right next to a change point and incur a localisation error of $\varepsilon n$.  When $\varepsilon = \Theta(1)$, as we have shown in Lemma 2, under the dynamic Huber contamination framework, the localisation error can still be of the order $\kappa^{-2} \sigma^2$, up to a logarithmic factor; while in the $O\cup I$ framework, the localisation error is at least of order $n$, which suggests inconsistency in change point localisation in the sense of (3). Due to its significant impact on the localisation error rate, we plan to leave the investigation under the $O\cup I$ framework as a future direction.
>
> __Point 4__: on the phase transition phenomenon. _[Has the transition phase phenomenon been also verified empirically?]_
>
> __Answer__: Thank you for your question.  Since the phase transition includes unknown constants, i.e.~the transition boundary is only rate optimal, it is not possible based on our current knowledge to conduct thorough empirical verification.  Loosely speaking, Figure 3 shows that when $\varepsilon$ is large (the right picture in each pair), the mean errors of both the number of detected change points and localisation error increase for all algorithms.  Although our algorithm still maintains a reasonable performance when $\varepsilon = 0.2$ with the configuration in Figure 3, further increasing $\varepsilon$ or decreasing the signal $\kappa$ will fall into the regime where it is impossible to consistently detect and localise the change points.
>
> __Point 6__: on the case $\kappa = 0$. [...$\kappa = 0$ only means that the smallest gap is 0, not that there is no change point, no?]
>
> __Answer__:  You are right that even when $\kappa = 0$, there might still be change points, but they are not of our interest. In this paper, as described in Assumption 1, we are interested in detecting change points in the means of the inlier distributions $F_i$'s. So when $\kappa = 0$, there is no change point of interest, but there may still be unwelcome change points due to the adversarial contamination.  This imposes a greater challenge and our algorithm will desirably output no change point of interest with high probability in line with the spirit of robustness literature.

---

> ### Author Response · Authors · 2021-08-31
> **Responses to the updates**
>
> We thank the reviewer for the further suggestions.  We would endeavour to include more explanations based on your suggestions in the revision.

---

### Decision · Program_Chairs · 2021-09-28

**Decision:**

Accept (Poster)

**Comment:**

The submission characterize a phase transition in a change-point univariate problem where a fraction of the time points
are contaminated by adversarially chosen distributions. The problem is clearly posed and the phase transition
had not been previously reported.

There is strong enthusiasm among most referees that the contributions are significant.
While some referees suggested interesting extensions to multivariate data and other forms of adversarial contamination,
the univariate phase transition under Huber's contamination model discovered in
the current submission is a sufficiently solid and complete contribution on its own.
Based on this, leaving these extensions for future work is reasonable and I recommend to accept the paper.


**Consistency Experiment:**

NeurIPS has a long history of experimentation. In 2014, NeurIPS ran an experiment in which 10% of submissions were reviewed by two independent committees to quantify the randomness in the review process. This year, we repeated a variant of this experiment to see how the quality of the review process has changed over time.  This paper was part of the experiment and was therefore assigned to two committees (consisting of reviewers, an Area Chair, and a Senior Area Chair) that reached independent decisions.  If both committees made the same recommendation, this recommendation was followed. If a single committee recommended acceptance, the paper was accepted (with the exception of a few cases in which the other committee identified what we considered a fatal flaw, e.g., an error in a key result).

Both committees reached the same decision: **Accept (Poster)**

The other committee assigned to the paper recommended **Accept (Poster)**.  You can find the other set of reviews, along with any follow up discussion with the authors here:
https://openreview.net/forum?id=xmMHxfE1qS6